# Improved Slime Mold Algorithm with Dynamic Quantum Rotation Gate and Opposition-Based Learning for Global Optimization and Engineering Design Problems

**Yunyang Zhang [1], Shiyu Du [2],\* and Quan Zhang [1]**

1   College of Information Science and Engineering, Ningbo University, Ningbo 315211, China
2   Engineering Laboratory of Advanced Energy Materials, Ningbo Institute of Materials Technology and Engineering, Ningbo 315211, China
\*   Correspondence: dushiyu@nimte.ac.cn

**Abstract:**   The slime mold algorithm (SMA) is a swarm-based metaheuristic algorithm inspired by the natural oscillatory patterns of slime molds. Compared with other algorithms, the SMA is competitive but still suffers from unbalanced development and exploration and the tendency to fall into local optima. To overcome these drawbacks, an improved SMA with a dynamic quantum rotation gate and opposition-based learning (DQOBLSMA) is proposed in this paper. Specifically, for the first time, two mechanisms are used simultaneously to improve the robustness of the original SMA: the dynamic quantum rotation gate and opposition-based learning. The dynamic quantum rotation gate proposes an adaptive parameter control strategy based on the fitness to achieve a balance between exploitation and exploration compared to the original quantum rotation gate. The opposition-based learning strategy enhances population diversity and avoids falling into the local optima. Twenty-three benchmark test functions verify the superiority of the DQOBLSMA. Three typical engineering design problems demonstrate the ability of the DQOBLSMA to solve practical problems. Experimental results show that the proposed algorithm outperforms other comparative algorithms in convergence speed, convergence accuracy, and reliability.

**Keywords:** slime mold algorithm; metaheuristics algorithm; engineering design problem; dynamic quantum rotation gate; opposition-based learning

## 1. Introduction

In the optimization field, solving an optimization problem usually means finding the optimal value to maximize or minimize a set of objective functions without violating constraints [1]. Optimization methods can be divided into two main categories: exact algorithms and metaheuristics [2]. While exact algorithms can provide global optima precisely, they have exponentially increasing execution times in proportion to the number of variables and are considered less suitable and practical [3]. In contrast, metaheuristic algorithms can identify the best or near-optimal solution in a reasonable amount of time [4]. During the last two decades, metaheuristic algorithms have gained much attention, and much development and work there have been on them due to their flexibility, simplicity, and global optimization. Thus, they are widely used for solving optimization problems in almost every domain, such as big data text clustering [5], tuning of fuzzy control systems [6,7], path planning [8,9], feature selection [10–12], training neural networks [13], parameter estimation for photovoltaic cells [14–16], image segmentation [17,18], tomography analysis [19], and permutation flowshop scheduling [20,21].

Metaheuristic algorithms simulate natural phenomena or laws of physics and are usually classified into three categories: evolutionary algorithms, physical and chemical algorithms, and swarm-based algorithms. Evolutionary algorithms are a class of algorithms that simulate the laws of evolution in nature. The best known is the genetic algorithm

(GA) [22], which was developed from Darwin's theory of superiority and inferiority. There are other algorithms, such as differential evolution (DE) [23], which simulates the crossover and variation mechanisms of inheritance, evolutionary programming (EP) [24], and evolutionary strategies (ES) [25]. Physical and chemical algorithms search for the optimum by simulating the universe's chemical laws or physical phenomena. Algorithms in this category include simulated annealing (SA) [26], electromagnetic field optimization (EFO) [27], equilibrium optimizer (EO) [28], and Archimedes' optimization algorithm (ArchOA) [29]. Swarm-based algorithms simulate the behavior of social groups of animals or humans. Examples of such algorithms include the whale optimization algorithm (WOA) [30], salp swarm algorithm (SSA) [31], moth search algorithm (MSA) [32], aquila optimizer (AO) [33], grey wolf optimizer (GWO) [34], harris hawks optimization (HHO) [35], and particle swarm optimization (PSO) [36].

However, the no free lunch (NFL) theorem [37] proves that no single algorithm can solve all optimization problems well. If an algorithm is particularly effective for a particular class of problems, it may not be able to solve other classes of optimization problems. This motivates us to propose new algorithms or improve the existing ones. The slime mold algorithm (SMA) [38] is a new meta-heuristic algorithm proposed by Li et al. in 2020. The basic idea of the SMA is based on the foraging behavior of slime mold, which has different feedback aspects according to the food quality. Different search mechanisms have been introduced into the SMA to solve various optimization problems. For example, Zhao et al. [39] introduced a diffusion mechanism and association strategy into the SMA and applied the proposed algorithm to the image segmentation of CT images. Salah L. et al. [40] applied the slime mold algorithm to optimize an artificial neural network model for predicting monthly stochastic urban water demand. Wang et al. [41] developed a parallel slime mold algorithm for the distribution network reconfiguration problem with distributed generation. Tang et al. [42] introduced chaotic opposition-based learning and spiral search strategies into the SMA and proposed two adaptive parameter control strategies. The simulation results show that the proposed algorithms outperform other similar algorithms. Örnek et al. [43] proposed an enhanced SMA that combines the sine cosine algorithm with the position update of the SMA. Experimental results show that the proposed hybrid algorithm has a better ability to jump out of local optima with faster convergence.

Although the SMA, as a new algorithm, is competitive with other algorithms, it also suffers from some shortcomings. The SMA, similarly to many other swarm-based metaheuristic algorithms, suffers from slow convergence and premature convergence to a local optimum solution [44]. In addition, the update strategy of SMA reduces exploration capabilities and reduces population diversity. To improve the above problems, an improved algorithm based on SMA, called the dynamic-quantum-rotation-gate- and opposition-based learning SMA (DQOBLSMA), is proposed. In this paper, we introduce two mechanisms, the dynamic quantum rotation gate (DQGR) and opposition-based learning (OBL), into the SMA simultaneously. Both mechanisms improve the shortcomings of the original algorithm in terms of slow convergence and the tendency to fall into local optima. First, DQGR rotates the search individuals to the direction of the optimum, improving the diversity of the population and enhancing the global exploration capability of the algorithm. At the same time, OBL explores the partial solution in the opposite direction, improving the algorithm's ability to jump out of local optima. The performance of the DQOBLSMA was evaluated by comparing it with the original SMA algorithm and with other advanced algorithms. In addition, three different constraint engineering problems were used to verify the performance of the DQOBLSMA further: the welded beam design problem, the tension/compression spring design problem, and pressure vessel design.

The main contributions of this paper are summarized as follows:

1. DQRG and OBL strategies were introduced into SMA to improve the exploration capabilities of SMA.
2. The DQRG strategy is proposed in order to balance the exploration and exploitation phases.

3. By comparing five well-known metaheuristic algorithms, experiments show that the proposed DQOBLSMA is more robust and effective.
4. Experiments on three engineering design optimization problems show that the DQOBLSMA can be effectively applied to practical engineering problems.

This paper is organized as follows. Section 2 describes the slime mold algorithm, quantum rotation gate, and opposition-based learning. Section 3 presents the proposed improved slime mold algorithm. Sections 4 show the experimental study and discussion using benchmark functions. The DQOBLSMA is applied to solve the three engineering problems in Section 5. Finally, the conclusion and future work are given in Section 6.

## 2. Materials and Methods

### 2.1. Slime Mold Algorithm

The slime mold algorithm (SMA) [38] is a swarm-based metaheuristic algorithm recently developed by Li et al. The algorithm simulates a range of behaviors for foraging by the slime mold. The negative and positive feedbacks of the slime mold using a biological oscillator to propagate waves during foraging for a food source are simulated by the SMA using adaptive weights. Three special behaviors of the slime mold are mathematically formulated in the SMA: approaching food, wrapping food, and grabbing food. The process of approaching food can be expressed as

$$Xi(t+1) = \begin{cases} Xb(t) + vb \cdot (W \cdot XA(t) - XB(t)), & r < p \\ vc \cdot Xi(t), & r \geq p \end{cases} \tag{1}$$

where $t$ is the number of current iterations, $X_i(t+1)$ is the newly generated position, $Xb(t)$ denotes the best position found by the slime mold in iteration $t$, $X_A(t)$ and $X_B(t)$ are two random positions selected from the population of slime mold, and $r$ is a random value in $[0, 1]$.

$vb$ and $vs.c$ are the coefficients that simulate the oscillation and contraction mode of slime mold, respectively, and $vs.c$ is designed to linearly decrease from one to zero during the iterations. The range of $vb$ is from $-a$ to $a$, and the computational formula of $a$ is

$$a = \operatorname{arctanh}\left(1 - \frac{t}{T}\right) \tag{2}$$

where $T$ is the maximum number of iterations.

According to Equations (1) and (2), it can be seen that as the number of iterations increases, the slime mold will wrap the food.

$W$ is a significantly important factor that indicates the weight of the slime mold, and it is calculated as follows:

$$W(\text{SmellIndex}(i)) = \begin{cases} 1 + \text{rand} \cdot \log\left(\frac{bF - S(i)}{bF - wF} + 1\right), i \leq N/2 \\ 1 - \text{rand} \cdot \log\left(\frac{bF - S(i)}{bF - wF} + 1\right), i > N/2 \end{cases} \tag{3}$$

$$SmellIndex(i) = Sort(S(i)) \tag{4}$$

where $N$ is the size of the population, $i$ represents the $i - th$ individual in the population, $i \in 1, 2 \ldots N$, $rand$ denotes the random value in the interval of $[0, 1]$, $bF$ denotes the optimal fitness obtained in the current iterative process, $wF$ denotes the worst fitness value obtained in the iterative process currently, $S(i)$ represents the fitness of $X$, $SmellIndex$ denotes the sequence of fitness values sorted.

$$p = \tanh|S(i) - DF| \tag{5}$$

where $DF$ denotes the best fitness obtained in all iterations.

Finally, when the slime mold has found the food , it still has a certain chance $z$ to search other new food, which is formulated as

$$X(t+1) = \text{rand} \cdot (UB - LB) + LB, r_2 < z \tag{6}$$

where $UB$ and $LB$ are the upper and lower limits, respectively, and $r_2$ implies a random value in the region $[0, 1]$. $z$ is set to 0.03 in original SMA.

Finally, the pseudo-code of SMA is given in Algorithm 1.

---

**Algorithm 1:** Pseudo-code of the slime mold algorithm (SMA).

---

**Input:** Population size $N$, Maximum number of iteration $MaxIt$.
**Output:** The best location $X_b$, the best fitness value *bestFitness* .

1  Initialize the parameters popsize($N$);
2  Initialize the positions of slime mold $X_i (i = 1, 2, 3..., N)$ ;
3  **while** $t < MaxIt$ **do**
4       Calculate the fitness of all slime molds;
5       Update *bestFitness*,$X_b$;
6       Calculate the $W$ by Equation (3);
7       **foreach** *each slime mold* **do**
8           **if** $r_2 < z$ **then**
9               update the position using Equation (6);
10          **else**
11              Update $p$, $vb$, and $vc$;
12              Update position by Equation (1);
13          **end**
14      **end**
15      $t = t + 1$;
16 **end**
17 **return** *bestFitness*,$X_b$

---

### 2.2. Description of the Quantum Rotation Gate

#### 2.2.1. Quantum Bit

The fundamental storage unit is a quantum bit in quantum computer systems, communication systems, and other quantum information systems [45]. The difference between quantum bits and classical bits is that quantum bits can be in a superposition of two states simultaneously, whereas classical bits can be in only one state at a period of time, which is defined as Equation (7).

$$|\phi\rangle = \alpha|0\rangle + \beta|1\rangle \tag{7}$$

where $\alpha$ and $\beta$ represent the probability amplitudes of the two superposition states. $|\alpha|^2$ and $|\beta|^2$ are the e probabilities that the qubit is in two different states of "0" and "1", and the relationship between them is shown in Equation (8).

$$|\alpha|^2 + |\beta|^2 = 1 \tag{8}$$

Thus, a quantum bit can represent one state or be in both states at the same time.

#### 2.2.2. Quantum Rotation Gate

In the DQOBLSMA, the QRG strategy is introduced to update the position of some search individuals to enhance the exploitation of the algorithm. In the physical discipline of quantum computing, the quantum rotation gate is used as a state processing technique. Quantum bits are binary, and the position information generated by the swarm-based algorithm is floating-point data. In order to process the position information, the discrete data of the quantum bits need to be turned into the algorithm's continuous data. The

information of each dimension of the search agent is rotated in couples and updated by a quantum rotation gate. The update process and adjustment operation of QRG are as follows. Equation (9) shows that the $2 \times 2$ matrix represents the quantum rotation gate.

$$U(\theta_i) = \begin{bmatrix} \cos(\theta_i) & -\sin(\theta_i) \\ \sin(\theta_i) & \cos(\theta_i) \end{bmatrix} \tag{9}$$

The updating process is as follows:

$$\begin{bmatrix} \alpha'_i \\ \beta'_i \end{bmatrix} = U(\theta_i) \begin{bmatrix} \alpha_i \\ \beta_i \end{bmatrix} = \begin{bmatrix} \cos(\theta_i) & -\sin(\theta_i) \\ \sin(\theta_i) & \cos(\theta_i) \end{bmatrix} \begin{bmatrix} \alpha_i \\ \beta_i \end{bmatrix} \tag{10}$$

where $(\alpha_i, \beta_i)^T$ shows the state of the quantum bit of the $i$th quantum bit of the chromosome before the update of the quantum rotation gate, and $(\alpha'_i, \beta'_i)^T$ indicates the state of the quantum bit after the update. $\theta_i$ denotes the rotation angle of the ith quantum bit, the size and sign of which have been pre-set, and its adjustment strategy is shown in Table 1.

**Table 1.** Strategies for specifying rotation angle in QRG.

| Situation | $\Delta\theta_i$ | $s(\alpha_i, \beta_i)$ | | | |
|---|---|---|---|---|---|
| | | $\alpha_i \beta_i < 0$ | $\alpha_i = 0$ | $\alpha_i \beta_i > 0$ | $\beta_i = 0$ |
| $f(x_i) = best\_fitness$ | $\delta$ | 0 | 0 | 0 | 0 |
| $f(x_i) > best\_fitness$ | $\delta$ | $-1$ | $\pm 1$ | $+1$ | 0 |
| $f(x_i) < best\_fitness$ | $\delta$ | $+1$ | 0 | $-1$ | $\pm 1$ |

Table 1 shows the rotation angle is labeled by $\theta_i = \Delta\theta_i \cdot s(\alpha_i, \beta_i)$, where $s(\alpha_i, \beta_i)$ denotes the rotation of the target direction. $\Delta\theta_i$ represents the rotation angle of the $i$-th rotation, where the position state of the $i$-th search agent in the population is $\alpha_i$, and the position state of the optimal search agent in the whole population is $\beta_i$. By comparing the fitness values of the current target and the optimal target, the direction of the target with higher fitness is selected to rotate the individual, thereby expanding the search space. If $f(x_i) > best\_fitness$, then the algorithm evolves toward the current target. Conversely, let the quantum bit state vector transform towards the direction where the optimal individual exists [46]. Figure 1 shows the quantum bit state vector transformation process.

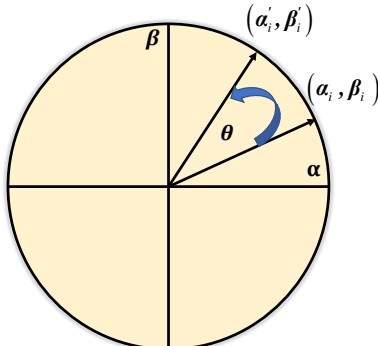

**Figure 1.** The process of updating the state of a quantum bit.

### 2.3. Opposition-Based Learning (OBL)

Tizhoosh proposed OBL in 2005 [47]. This technique can increase the convergence speeds of metaheuristic algorithms by replacing a solution in the population by searching for a potentially better solution in the opposite direction of the current one. With this approach, a population with better solutions could be generated after each iteration and accelerate convergence speed. The OBL strategy has been successfully used in various

metaheuristic algorithms to improve the ability of local optima stagnation avoidance [48], and the mathematical expression is as follows:

$$X_{\text{OBL}}(t) = LB + UB - X(t) \tag{11}$$

In opposition-based learning, for the original solution $X(t)$ and the reverse solution $X_{\text{OBL}}(t)$, according to their fitness, save the better solution among them. Finally, the slime mold position for the next iteration is updated as follows in the minimization problem:

$$X_{\text{OBL}}(t+1) = \begin{cases} X_{\text{OBL}}(t) & \text{if } f(X_{\text{OBL}}(t)) < f(X(t)) \\ X(t) & \text{if } f(X_{\text{OBL}}(t)) \geq f(X(t)) \end{cases} \tag{12}$$

## 3. Proposed Method

### 3.1. Improved Quantum Rotation Gate

The magnitude of the rotation angle of the QRG significantly affects the convergence speed. A relatively large amplitude leads to premature convergence. Conversely, smaller angles lead to slower convergence. In particular, the rotation angle of the original quantum rotation gate is fixed, which is not conducive to the balance between exploration and exploitation. Based on this, we propose a new dynamic adaptation strategy to adjust the rotation angle of the quantum rotation gate. In the early exploration stage, the value of $\theta$ should be increased when the current individual is far from the best. In the exploitation stage, the value of $\theta$ should be decreased. This method allows the search process to adapt to different solutions and is more conducive to searching for the global optimum. In detail, this improved method determines the value of the rotation angle by the difference between the current individual's fitness and the best fitness that has been obtained so far. The rotation angle $\theta$ is defined as

$$\Delta\theta = \theta_{\min} + \gamma_i \cdot (\theta_{\max} - \theta_{\min}) \tag{13}$$

where $\theta_{\max}$ and $\theta_{\min}$ are the maximum and minimum values of the range of $\Delta\theta$, respectively. The maximum and minimum values take $0.035\pi$ and $0.001\pi$, respectively. $\gamma$ is defined as:

$$\gamma_i = 1 - e^{-4 \cdot \left(\frac{bF - S(i)}{bF - wF}\right)^2} \tag{14}$$

The pseudo-code of DQRG (Algorithm 2) is as follows:

---
**Algorithm 2:** Pseudo-code of the quantum rotation gate (DQRG).

---
**Input:** position $X_i$, fitness values of $X_i$ $fitness(i)$, the best fitness value $bF$, $dim$
**Output:** updated position $X$.

1 Initialize the parameters $\alpha$, $\beta$, $s$;
2 **while** $i < dim$ **do**
3     Update $\alpha$, $\beta$;
4     Compare the $fitness(i)$ and $bF$;
5     Update $s$ according to Table 1;
6     Update $\Delta\theta$ by Equation (13);
7     Perform DQRG by Equation (10);
8     $i = i + 1$;
9 **end**
10 **return** $X$

---

### 3.2. OBL

In this work, an improved method to obtain the opposite solution is proposed further. Specifically, instead of using only lower and upper bounds to find the opposite point, the impact of the current better solution, including the optimal, suboptimal, and third optimal

solutions, is added to the opposite point's calculation procedure. The new formula of the opposite point is expressed as follows:

$$X_m = \frac{X_{os} + X_{ss} + X_{ts}}{3} \tag{15}$$

where $X_m$ is the average of three better solutions, $X_{os}$ is the current best solution, $X_{ss}$ is the suboptimal solution, and $X_{ts}$ is the third optimal solution.

$$X_{\text{OBL}}(t+1) = LB + UB - X_m(t) + rand \cdot (X_m(t) - X(t)) \tag{16}$$

where $X_{\text{OBL}}(t+1)$ is the improved opposite solution, *rand* denotes the random value in the interval of $[0,1]$, and *UB* and *LB* are the upper and lower limits.

*3.3. Improved SMA*

To explore the solution space of complex optimization problems more efficiently, we propose two strategies based on the original SMA algorithm: the DQRG and OBL strategies. In the proposed method, two main conditions are considered to execute the proposed policy procedures. The first condition is the execution of SMA or two other strategies. If $r_2 < 0.8$, then SMA is executed to update the position. Otherwise, the second condition is checked to determine the strategy to adopt. If $r_3 < 0.5$ in the second condition, the solution will be updated using the DQRG; otherwise, OBL will be executed for the searched individual. The pseudo-code of the DQOBLSMA is shown as Algorithm 3:

---
**Algorithm 3:** Pseudo-code of the DQOBLSMA.

---
**Input:** Population size $N$, Maximum number of iteration *MaxIt*.
**Output:** The best location $X_b$, the best fitness value $bF$ .

1 Initialize the parameters popsize($N$), $Max_i teraition$;
2 Initialize the positions of slime mold $X_i (i = 1, 2, 3..., N)$ ;
3 **while** $t < MaxIt$ **do**
4      Calculate the fitness of all slime mold;
5      Update $bF$, $X_b$;
6      Calculate the $W$ by Equation (3);
7      **foreach** *slime mold* **do**
8          **if** $r_1 < 0.8$ **then**
9              **if** $r_2 < z$ **then**
10                  update the position using Equation (6);
11              **else**
12                  Update $p$, $vb$, and $vc$;
13                  Update position by Equation (1);
14              **end**
15          **else**
16              **if** $r_3 < 0.5$ **then**
17                  Operate Dynamic quantum rotation gate by Algorithm 2;
18              **else**
19                  Calculate opposition solution $X_{OBL}$ of individual $X$ by Equation (16);
20              **end**
21          **end**
22      **end**
23      $t = t + 1$;
24 **end**
25 **return** *bestFitness*, $X_b$

---

### 3.4. Computational Complexity Analysis

The computational complexity of the DQOBLSMA depends on the population size ($N$), dimension size ($D$), and maximum iterations ($T$). First, the DQOBLSMA produces the search agents randomly in the search space, so the computational complexity is $O(N \times D)$. Second, the computational complexity of calculating the fitness of all agents is $O(N)$. The quick-sort of all search agents is $O(N \times \log N)$. Moreover, updating the positions of agents in the original SMA is $(N \times D)$. Therefore, the total computational complexity of original SMA is $O(N \times D + N \times T \times (1 + D + \log N))$.

Updating the positions through the DQRG is $O(N \times D)$ (maximum), and the OBL is $O(N)$ (maximum). Updating the position using DQRG and the original SMA will not be done simultaneously. In summary, the final time complexity is $O(DQOBLSMA) = O(N \times D + N \times T \times (1 + D + \log N))$(maximum). In summary, the improved strategy proposed in this paper does not increase the computational complexity when compared with the original SMA.

## 4. Experiments and Discussion

We conducted a series of experiments to verify the performance of the DQOBLSMA. The classical benchmark functions are introduced in Section 4.1. In the experiments of test functions, the impacts of two mechanisms were analyzed; see Section 4.2. In Section 4.3, the DQOBLSMA is compared with several advanced algorithms. In Section 4.4, the convergence of the algorithms is analyzed.

The performance of the DQOBLSMA was investigated using the mean result (Mean) and standard deviation (Std). In order to accurately make statistically reasonable conclusions, the results of the benchmark test functions were ranked using the Friedman test. In addition, the Wilcoxon's rank-sum test was used to assess the average performances of the algorithms in a statistical sense. In this study, it was used to test whether there was a difference in the effect of the DQOBLSMA compared with those of other algorithms in pairwise comparisons. When the p-value is less than 0.05, the result is significantly different from the other methods. The symbols "+," "−," and "=" indicate if the DQOBLSMA is better than, inferior to, or equal to the other algorithms, respectively.

### 4.1. Benchmark Function Validation and Parameter Settings

In this study, the test set for the DQOBLSMA comparison experiment was the 23 classical test functions that had been used in the literature [34]. The details are shown in Table 2. These classical test functions are divided into unimodal functions, multimodal functions, and fixed-dimension multimodal functions. The unimodal functions (F1–F7) have only one local solution and one optimal global solution and are usually used to evaluate the local exploitation ability of the algorithm. Multimodal functions (F8–F13) are often used to test the exploration ability of the algorithm. F14–F23 are fixed-dimensional multimodal functions with many local optimal points and low dimensionality, which can be used to evaluate the stability of the algorithm.

**Table 2.** The classic benchmark functions.

| Function Type | Function | Name | Dimension | Range | Theoretical Value |
|---|---|---|---|---|---|
| Unimodal test functions | F1 | Sphere | 30 | $[-100, 100]$ | 0 |
| | F2 | Schwefel 2.22 | 30 | $[-10, 10]$ | 0 |
| | F3 | Schwefel 1.2 | 30 | $[-100, 100]$ | 0 |
| | F4 | Schwefel 2.21 | 30 | $[-100, 100]$ | 0 |
| | F5 | Rosenbrock | 30 | $[-30, 30]$ | 0 |
| | F6 | Step | 30 | $[-100, 100]$ | 0 |
| Multimodal test functions | F7 | Quartic | 30 | $[-1.28, 1.28]$ | 0 |
| | F8 | Schwefel 2.26 | 30 | $[-500, 500]$ | $-418.9829 \times D$ |
| | F9 | Rastrigin | 30 | $[-5.12, 5.12]$ | 0 |
| | F10 | Ackley | 30 | $[-32, 32]$ | 0 |
| | F11 | Griewank | 30 | $[-600, 600]$ | 0 |
| | F12 | Penalized | 30 | $[-50, 50]$ | 0 |
| | F13 | Penalized2 | 30 | $[-50, 50]$ | 0 |
| Fixed-dimension multimodal test functions | F14 | Foxholes | 2 | $[-65, 65]$ | 0.998004 |
| | F15 | Kowalik | 4 | $[-5, 5]$ | 0.0003075 |
| | F16 | Six-Hump Camel Back | 2 | $[-5, 5]$ | $-1.03163$ |
| | F17 | Branin | 2 | $[-5, 5]$ | 0.398 |
| | F18 | Goldstein Price | 2 | $[-2, 2]$ | 3 |
| | F19 | Hartman 3 | 3 | $[-1, 2]$ | $-3.8628$ |
| | F20 | Hartman 6 | 6 | $[0, 1]$ | $-3.322$ |
| | F21 | Shekel 5 | 4 | $[0, 10]$ | $-10.1532$ |
| | F22 | Shekel 7 | 4 | $[0, 10]$ | $-10.4028$ |
| | F23 | Shekel 10 | 4 | $[0, 10]$ | $-10.5363$ |

The DQOBLSMA has been compared to the original SMA and five other algorithms: the slime mold algorithm improved by opposition-based learning and Levy flight distribution (OBLSMAL) [48], the equilibrium slime mold algorithm (ESMA) [49], the equilibrium optimizer with a mutation strategy (MEO) [50], the adaptive differential evolution with an optional external archive (JADE) [51], and the gray wolf optimizer based on random walk (RWGWO) [52]. The parameter settings of each algorithm are shown in Table 3, and the experimental parameters for all optimization algorithms were chosen to be the same as those reported in the original works.

**Table 3.** Parameter settings for the comparative algorithms.

| Algorithm | Parameter |
|---|---|
| OBLSMAL | $z = 0.03, p_1 = 0.5, p_2 = 0.5$ |
| ESMA | $z = 0.03$ |
| MEO | $a_1 = 2, a_2 = 1, GP = 0.5$ |
| JADE | $\mu F = 0.5, \mu CR = 0.5, p = 0.1, c = 0.1$ |
| RWGWO | Control parameter $a, b$ decrease linearly from 2 to 0 |
| SMA | $z = 0.03$ |

In order to maintain a fair comparison, each algorithm was independently run 30 times. The population size ($N$) and the maximum function evaluation times ($FEs$) of all experimental methods were fixed at 30 and 15,000, respectively. The comparative experiment was run under the same test conditions to keep the experimental conditions consistent. The proposed method was coded in Python3.8 and tested on a PC with an AMD R5-4600 Hz, 3.00 GHz of memory, 16 GB of RAM, and the Windows 11 operating system.

### 4.2. Impacts of Components

In this section, different versions of the improvement are investigated. The proposed DQOBLSMA adds two different mechanisms to the original SMA. To verify their respective effects, they are compared when separated. Different combinations between SMA and two mechanisms are listed below:

- SMA combined with DQRG and OBL (DQOBLSMA);
- SMA combined with DQRG (DQSMA);
- SMA combined with OBL(OBLSMA);
- Original SMA;

Table 4 gives the comparison results between the original SMA and the improved algorithm after adding the mechanism. The ranking of the four algorithms is given at the end of the table, and it can be seen that the first-ranked algorithm is the DQOBLSMA. This ranking was obtained using the Friedman ranking test [53] and reveals the overall performance rankings of the compared algorithms against the tested functions. In these cases, the ranking from best to worst was roughly as follows: DQOBLSMA > OBLSMA > SMA > DQSMA. With the addition of both mechanisms, the performance of the DQOBLSMA is more stable, and the global search capability is much improved. When comparing DQSMA with OBLSMA, we can see that OBLSMA is much stronger than DQSMA, indicating that the contribution of OBL to the performance of SMA is more significant than the contribution of DQRG to the performance of SMA. When comparing DQSMA with SMA, we can see that DQSMA becomes worse on unimodal functions but stronger on most multimodal and fixed-dimensional multimodal functions than the original SMA in terms of optimization.

Wilcoxon's rank-sum test was used to verify the significance of the DQOBLSMA against the original SMA and SMA with the addition of one mechanism. The results are shown in Table 5. Based on these results and those in Table 4, the DQOBLSMA outperformed SMA on 13 benchmark functions, DQSMA on 17 benchmark functions, and OBLSMA on 8 benchmark functions. Thus, the DQOBLSMA algorithm proposed in this paper combines DQRG with OBL. Although DQSMA and OBLSMA can both find the solutions, there are more benefits to be gained by combining the two strategies. In conclusion, the DQOBLSMA offers better optimization performance and is significantly better than SMA, DQSMA, and OBLSMA.

**Table 4.** Search results (comparisons of the DQOBLSMA, DQSMA, OBLSMA, SMA).

| Function | DQOBLSMA | | DQSMA | | OBLSMA | | SMA | |
|---|---|---|---|---|---|---|---|---|
| | **Mean** | **Std** | **Mean** | **Std** | **Mean** | **Std** | **Mean** | **Std** |
| F1 | $0.0000 \times 10^{+00}$ | $0.0000 \times 10^{+00}$ | $1.0891 \times 10^{-02}$ | $4.3266 \times 10^{-03}$ | $0.0000 \times 10^{+00}$ | $0.0000 \times 10^{+00}$ | $0.0000 \times 10^{+00}$ | $0.0000 \times 10^{+00}$ |
| F2 | $2.9368 \times 10^{-231}$ | $0.0000 \times 10^{+00}$ | $5.1658 \times 10^{-02}$ | $1.7908 \times 10^{-02}$ | $2.7971 \times 10^{-244}$ | $0.0000 \times 10^{+00}$ | $7.2130 \times 10^{-164}$ | $0.0000 \times 10^{+00}$ |
| F3 | $0.0000 \times 10^{+00}$ | $0.0000 \times 10^{+00}$ | $3.8217 \times 10^{-02}$ | $5.9525 \times 10^{-02}$ | $0.0000 \times 10^{+00}$ | $0.0000 \times 10^{+00}$ | $0.0000 \times 10^{+00}$ | $0.0000 \times 10^{+00}$ |
| F4 | $1.4919 \times 10^{-224}$ | $0.0000 \times 10^{+00}$ | $1.5620 \times 10^{-02}$ | $8.3506 \times 10^{-03}$ | $3.1204 \times 10^{-229}$ | $0.0000 \times 10^{+00}$ | $5.3508 \times 10^{-168}$ | $0.0000 \times 10^{+00}$ |
| F5 | $1.4718 \times 10^{-01}$ | $1.5834 \times 10^{-01}$ | $5.1129 \times 10^{+00}$ | $1.1125 \times 10^{+01}$ | $6.4059 \times 10^{+00}$ | $1.1204 \times 10^{+01}$ | $2.8202 \times 10^{+01}$ | $2.6986 \times 10^{-01}$ |
| F6 | $0.0000 \times 10^{+00}$ | $0.0000 \times 10^{+00}$ | $0.0000 \times 10^{+00}$ | $0.0000 \times 10^{+00}$ | $0.0000 \times 10^{+00}$ | $0.0000 \times 10^{+00}$ | $0.0000 \times 10^{+00}$ | $0.0000 \times 10^{+00}$ |
| F7 | $8.8202 \times 10^{-05}$ | $7.2479 \times 10^{-05}$ | $5.8003 \times 10^{-04}$ | $3.1965 \times 10^{-04}$ | $1.3372 \times 10^{-04}$ | $8.9724 \times 10^{-05}$ | $2.3852 \times 10^{-04}$ | $2.0182 \times 10^{-04}$ |
| F8 | $-1.2569 \times 10^{+04}$ | $1.0234 \times 10^{-01}$ | $-1.1726 \times 10^{+04}$ | $1.0829 \times 10^{+03}$ | $-1.2569 \times 10^{+04}$ | $5.6297 \times 10^{-02}$ | $-9.1620 \times 10^{+03}$ | $7.0236 \times 10^{+02}$ |
| F9 | $0.0000 \times 10^{+00}$ | $0.0000 \times 10^{+00}$ | $0.0000 \times 10^{+00}$ | $0.0000 \times 10^{+00}$ | $0.0000 \times 10^{+00}$ | $0.0000 \times 10^{+00}$ | $0.0000 \times 10^{+00}$ | $0.0000 \times 10^{+00}$ |
| F10 | $4.4409 \times 10^{-16}$ | $0.0000 \times 10^{+00}$ | $4.4409 \times 10^{-16}$ | $0.0000 \times 10^{+00}$ | $4.4409 \times 10^{-16}$ | $0.0000 \times 10^{+00}$ | $4.4409 \times 10^{-16}$ | $0.0000 \times 10^{+00}$ |
| F11 | $0.0000 \times 10^{+00}$ | $0.0000 \times 10^{+00}$ | $2.2635 \times 10^{-02}$ | $1.0138 \times 10^{-02}$ | $0.0000 \times 10^{+00}$ | $0.0000 \times 10^{+00}$ | $0.0000 \times 10^{+00}$ | $0.0000 \times 10^{+00}$ |
| F12 | $8.9207 \times 10^{-04}$ | $1.0608 \times 10^{-03}$ | $5.0820 \times 10^{-03}$ | $1.4513 \times 10^{-02}$ | $3.5431 \times 10^{-03}$ | $9.2857 \times 10^{-03}$ | $2.4763 \times 10^{-02}$ | $9.4810 \times 10^{-03}$ |
| F13 | $1.4921 \times 10^{-03}$ | $3.6813 \times 10^{-03}$ | $4.2575 \times 10^{-02}$ | $7.6342 \times 10^{-02}$ | $2.2321 \times 10^{-03}$ | $8.3069 \times 10^{-03}$ | $5.0605 \times 10^{-02}$ | $3.4525 \times 10^{-02}$ |
| F14 | $9.9800 \times 10^{-01}$ | $3.4807 \times 10^{-13}$ | $1.1634 \times 10^{+00}$ | $5.9405 \times 10^{-01}$ | $9.9800 \times 10^{-01}$ | $2.0372 \times 10^{-13}$ | $9.9800 \times 10^{-01}$ | $2.0923 \times 10^{-12}$ |
| F15 | $3.8029 \times 10^{-04}$ | $9.0820 \times 10^{-05}$ | $4.4014 \times 10^{-04}$ | $1.0909 \times 10^{-04}$ | $4.7568 \times 10^{-04}$ | $1.7299 \times 10^{-04}$ | $5.3389 \times 10^{-04}$ | $2.7098 \times 10^{-04}$ |
| F16 | $-1.0316 \times 10^{+00}$ | $4.1555 \times 10^{-10}$ | $-1.0316 \times 10^{+00}$ | $5.1500 \times 10^{-06}$ | $-1.0316 \times 10^{+00}$ | $8.8268 \times 10^{-10}$ | $-1.0316 \times 10^{+00}$ | $1.4953 \times 10^{-09}$ |
| F17 | $3.9789 \times 10^{-01}$ | $3.2851 \times 10^{-08}$ | $3.9794 \times 10^{-01}$ | $1.2457 \times 10^{-04}$ | $3.9789 \times 10^{-01}$ | $1.3247 \times 10^{-07}$ | $3.9789 \times 10^{-01}$ | $3.4597 \times 10^{-07}$ |
| F18 | $3.0000 \times 10^{+00}$ | $4.3415 \times 10^{-07}$ | $3.0006 \times 10^{+00}$ | $5.3604 \times 10^{-04}$ | $3.0000 \times 10^{+00}$ | $5.3937 \times 10^{-08}$ | $3.0000 \times 10^{+00}$ | $3.3742 \times 10^{-08}$ |
| F19 | $-3.8628 \times 10^{+00}$ | $6.0439 \times 10^{-07}$ | $-3.8628 \times 10^{+00}$ | $2.6271 \times 10^{-05}$ | $-3.8627 \times 10^{+00}$ | $3.4507 \times 10^{-04}$ | $-3.8628 \times 10^{+00}$ | $3.3028 \times 10^{-07}$ |
| F20 | $-3.2821 \times 10^{+00}$ | $5.7002 \times 10^{-02}$ | $-3.2375 \times 10^{+00}$ | $6.5585 \times 10^{-02}$ | $-3.2615 \times 10^{+00}$ | $6.0657 \times 10^{-02}$ | $-3.2582 \times 10^{+00}$ | $5.9773 \times 10^{-02}$ |
| F21 | $-1.0153 \times 10^{+01}$ | $2.1496 \times 10^{-04}$ | $-1.0152 \times 10^{+01}$ | $1.8979 \times 10^{-03}$ | $-1.0153 \times 10^{+01}$ | $8.5453 \times 10^{-05}$ | $-8.7668 \times 10^{+00}$ | $2.7426 \times 10^{+00}$ |
| F22 | $-1.0403 \times 10^{+01}$ | $1.8317 \times 10^{-04}$ | $-1.0402 \times 10^{+01}$ | $1.0712 \times 10^{-03}$ | $-1.0403 \times 10^{+01}$ | $1.2865 \times 10^{-04}$ | $-8.5645 \times 10^{+00}$ | $2.8449 \times 10^{+00}$ |
| F23 | $-1.0536 \times 10^{+01}$ | $2.0415 \times 10^{-04}$ | $-1.0534 \times 10^{+01}$ | $3.6030 \times 10^{-03}$ | $-1.0536 \times 10^{+01}$ | $1.2450 \times 10^{-04}$ | $-8.5593 \times 10^{+00}$ | $2.8800 \times 10^{+00}$ |
| Friedman test average rank | 1.74 | | 3.33 | | 1.91 | | 3.02 | |

**Table 5.** Test statistical results of Wilcoxon's rank-sum test.

| Benchmark | DQOBLSMA vs. DQSMA | | DQOBLSMA vs. OBLSMA | | DQOBLSMA vs. SMA | |
|---|---|---|---|---|---|---|
| | *p*-Value | Winner | *p*-Value | Winner | *p*-Value | Winner |
| F1 | $2.87 \times 10^{-11}$ | + | NaN | = | NaN | = |
| F2 | $2.87 \times 10^{-11}$ | + | $5.22 \times 10^{-09}$ | − | $1.94 \times 10^{-09}$ | + |
| F3 | $2.87 \times 10^{-11}$ | + | NaN | = | NaN | = |
| F4 | $2.87 \times 10^{-11}$ | + | $5.22 \times 10^{-09}$ | − | $1.48 \times 10^{-09}$ | + |
| F5 | NaN | + | $6.24 \times 10^{-03}$ | + | $2.87 \times 10^{-11}$ | + |
| F6 | NaN | = | NaN | = | NaN | = |
| F7 | $1.63 \times 10^{-08}$ | + | $2.37 \times 10^{-02}$ | + | $1.73 \times 10^{-04}$ | + |
| F8 | $2.87 \times 10^{-11}$ | + | $5.96 \times 10^{-03}$ | = | $2.87 \times 10^{-11}$ | + |
| F9 | $2.87 \times 10^{-11}$ | = | NaN | = | NaN | = |
| F10 | $2.87 \times 10^{-11}$ | = | NaN | = | $2.87 \times 10^{-11}$ | = |
| F11 | $2.87 \times 10^{-11}$ | + | NaN | = | NaN | = |
| F12 | NaN | + | $4.59 \times 10^{-02}$ | + | $2.87 \times 10^{-11}$ | + |
| F13 | $7.90 \times 10^{-05}$ | + | NaN | + | $3.88 \times 10^{-11}$ | + |
| F14 | $2.87 \times 10^{-11}$ | + | NaN | = | $2.87 \times 10^{-11}$ | = |
| F15 | $6.8 \times 10^{-3}$ | + | $3.09 \times 10^{-02}$ | + | $2.82 \times 10^{-03}$ | + |
| F16 | $2.87 \times 10^{-11}$ | = | NaN | = | $5.10 \times 10^{-05}$ | + |
| F17 | $2.87 \times 10^{-11}$ | + | NaN | = | $6.37 \times 10^{-04}$ | = |
| F18 | $2.87 \times 10^{-11}$ | = | NaN | + | NaN | = |
| F19 | $1.31 \times 10^{-07}$ | = | $5.12 \times 10^{-04}$ | + | $3.50 \times 10^{-08}$ | = |
| F20 | $1.15 \times 10^{-06}$ | + | NaN | + | $3.76 \times 10^{-03}$ | + |
| F21 | $6.81 \times 10^{-09}$ | + | $1.41 \times 10^{-03}$ | = | $2.33 \times 10^{-09}$ | + |
| F22 | $8.12 \times 10^{-09}$ | + | $4.44 \times 10^{-02}$ | = | $6.26 \times 10^{-08}$ | + |
| F23 | $1.54 \times 10^{-10}$ | + | $1.72 \times 10^{-03}$ | = | $1.55 \times 10^{-06}$ | + |
| +/−/= | 17/0/6 | | 8/2/13 | | f13/0/10 | |

### 4.3. Benchmark Function Experiments

As seen from Table 6, on unimodal benchmark functions (F1–F7), the DQOBLSMA can achieve better results than other optimization algorithms. For F1, F3, and F6, the DQOBLSMA could find the theoretical optimal value. For all unimodal functions, the DQOBLSMA obtained the smallest mean values and standard deviations compared to other algorithms, showing the best accuracy and stability.

From the results shown in Tables 7 and 8, the DQOBLSMA outperformed the other algorithms for most of the multimodal and fixed-dimensional multimodal functions. For the multimodal functions F8–F13, the DQOBLSMA obtained almost all the best mean and standard deviation values, and obtainedthe global optimal solution for four functions (F8–F11). As shown in Table 8, the DQOBLSMA obtained theoretically optimal values in 8 of the 10 fixed-dimensional multimodal functions (F14–F23). Although the DQOBLSMA did not outperform JADE in F14–F23, it exceeded ESMA and OBLSMAL in overall performance. These results show that the DQOBLSMA also provides powerful and robust exploitation capabilities.

In addition, Table 9 presents Wilcoxon's rank-sum test results to verify the significant differences between the DQOBLSMA and the other five algorithms. It is worth noting that *p*-values less than 0.05 mean significant differences between the respective pairs of compared algorithms. The DQOBLSMA outperformed all other algorithms to varying degrees, and outperformed OBLSMAL, ESMA, MEO, JADE, and RWGWO, on 14, 15, 16, 15, and 18 benchmark functions, respectively. Table 10 shows the statistical results of the Friedman test, where the DQOBLSMA ranked first in F1–F7 and F8–F13 and second after JADE by a small margin in F14–F23. The DQOBLSMA received the best ranking overall. In summary, the DQOBLSMA provided better results on almost all benchmark functions than the other algorithms.

**Table 6.** Results of unimodal benchmark test functions.

| Func | Criteria | DQOBLSMA | OBLSMAL | ESMA | MEO | JADE | RWGWO |
|------|----------|----------|---------|------|-----|------|-------|
| F1 | Best | $0.0000 \times 10^{+00}$ | $0.0000 \times 10^{+00}$ | $0.0000 \times 10^{+00}$ | $1.0936 \times 10^{-54}$ | $7.1160 \times 10^{-14}$ | $7.2435 \times 10^{-73}$ |
| | Mean | $0.0000 \times 10^{+00}$ | $0.0000 \times 10^{+00}$ | $0.0000 \times 10^{+00}$ | $1.3473 \times 10^{-51}$ | $1.3924 \times 10^{-12}$ | $9.9351 \times 10^{-65}$ |
| | Worst | $0.0000 \times 10^{+00}$ | $0.0000 \times 10^{+00}$ | $0.0000 \times 10^{+00}$ | $1.1718 \times 10^{-50}$ | $8.3623 \times 10^{-12}$ | $2.8903 \times 10^{-63}$ |
| | Std | $0.0000 \times 10^{+00}$ | $0.0000 \times 10^{+00}$ | $0.0000 \times 10^{+00}$ | $3.4983 \times 10^{-51}$ | $2.2303 \times 10^{-12}$ | $6.9913 \times 10^{-64}$ |
| F2 | Best | $1.6860 \times 10^{-280}$ | $1.8971 \times 10^{-126}$ | $1.2829 \times 10^{-179}$ | $1.2944 \times 10^{-32}$ | $8.7945 \times 10^{-08}$ | $8.4261 \times 10^{-52}$ |
| | Mean | $2.9368 \times 10^{-231}$ | $7.4709 \times 10^{-113}$ | $4.1210 \times 10^{-175}$ | $6.2425 \times 10^{-31}$ | $4.5037 \times 10^{-06}$ | $1.2077 \times 10^{-47}$ |
| | Worst | $8.8104 \times 10^{-230}$ | $2.1489 \times 10^{-111}$ | $8.3686 \times 10^{-174}$ | $2.2747 \times 10^{-30}$ | $7.1303 \times 10^{-05}$ | $7.7841 \times 10^{-47}$ |
| | Std | $0.0000 \times 10^{+00}$ | $5.1975 \times 10^{-112}$ | $0.0000 \times 10^{+00}$ | $7.6873 \times 10^{-31}$ | $1.7166 \times 10^{-05}$ | $2.3314 \times 10^{-47}$ |
| F3 | Best | $0.0000 \times 10^{+00}$ | $0.0000 \times 10^{+00}$ | $1.3923 \times 10^{-278}$ | $3.5006 \times 10^{-21}$ | $3.4830 \times 10^{+00}$ | $2.2232 \times 10^{+03}$ |
| | Mean | $0.0000 \times 10^{+00}$ | $0.0000 \times 10^{+00}$ | $7.5255 \times 10^{-205}$ | $1.7481 \times 10^{-17}$ | $2.1172 \times 10^{+01}$ | $6.0553 \times 10^{+03}$ |
| | Worst | $0.0000 \times 10^{+00}$ | $0.0000 \times 10^{+00}$ | $2.2576 \times 10^{-203}$ | $1.1762 \times 10^{-16}$ | $5.6794 \times 10^{+01}$ | $1.1356 \times 10^{+04}$ |
| | Std | $0.0000 \times 10^{+00}$ | $0.0000 \times 10^{+00}$ | $0.0000 \times 10^{+00}$ | $3.4991 \times 10^{-17}$ | $1.6585 \times 10^{+01}$ | $2.4311 \times 10^{+03}$ |
| F4 | Best | $2.2279 \times 10^{-273}$ | $9.2369 \times 10^{-122}$ | $8.7764 \times 10^{-173}$ | $6.6498 \times 10^{-15}$ | $1.2412 \times 10^{-01}$ | $8.3027 \times 10^{-07}$ |
| | Mean | $1.4919 \times 10^{-224}$ | $1.3337 \times 10^{-106}$ | $1.4870 \times 10^{-162}$ | $5.1881 \times 10^{-13}$ | $6.6358 \times 10^{-01}$ | $2.1528 \times 10^{+00}$ |
| | Worst | $4.4756 \times 10^{-223}$ | $3.9116 \times 10^{-105}$ | $4.2072 \times 10^{-161}$ | $5.2753 \times 10^{-12}$ | $1.7608 \times 10^{+00}$ | $2.9139 \times 10^{+01}$ |
| | Std | $0.0000 \times 10^{+00}$ | $9.4644 \times 10^{-106}$ | $1.0186 \times 10^{-161}$ | $1.2870 \times 10^{-12}$ | $4.2471 \times 10^{-01}$ | $7.2432 \times 10^{+00}$ |
| F5 | Best | $4.8100 \times 10^{-04}$ | $2.6149 \times 10^{+01}$ | $2.3534 \times 10^{+01}$ | $2.5670 \times 10^{+01}$ | $1.5204 \times 10^{+01}$ | $2.8626 \times 10^{+01}$ |
| | Mean | $1.4718 \times 10^{-01}$ | $2.7476 \times 10^{+01}$ | $2.7593 \times 10^{+01}$ | $2.6755 \times 10^{+01}$ | $3.4093 \times 10^{+01}$ | $2.8807 \times 10^{+01}$ |
| | Worst | $5.6207 \times 10^{-01}$ | $2.8866 \times 10^{+01}$ | $2.8973 \times 10^{+01}$ | $2.8759 \times 10^{+01}$ | $9.3404 \times 10^{+01}$ | $2.8898 \times 10^{+01}$ |
| | Std | $1.5834 \times 10^{-01}$ | $8.0237 \times 10^{-01}$ | $1.5217 \times 10^{+00}$ | $7.3628 \times 10^{-01}$ | $2.4394 \times 10^{+01}$ | $6.2979 \times 10^{-02}$ |
| F6 | Best | $0.0000 \times 10^{+00}$ | $0.0000 \times 10^{+00}$ | $0.0000 \times 10^{+00}$ | $0.0000 \times 10^{+00}$ | $0.0000 \times 10^{+00}$ | $0.0000 \times 10^{+00}$ |
| | Mean | $0.0000 \times 10^{+00}$ | $0.0000 \times 10^{+00}$ | $0.0000 \times 10^{+00}$ | $0.0000 \times 10^{+00}$ | $6.6667 \times 10^{-02}$ | $0.0000 \times 10^{+00}$ |
| | Worst | $0.0000 \times 10^{+00}$ | $0.0000 \times 10^{+00}$ | $0.0000 \times 10^{+00}$ | $0.0000 \times 10^{+00}$ | $1.0000 \times 10^{+00}$ | $0.0000 \times 10^{+00}$ |
| | Std | $0.0000 \times 10^{+00}$ | $0.0000 \times 10^{+00}$ | $0.0000 \times 10^{+00}$ | $0.0000 \times 10^{+00}$ | $2.9152 \times 10^{-01}$ | $0.0000 \times 10^{+00}$ |
| F7 | Best | $3.2865 \times 10^{-07}$ | $5.4366 \times 10^{-07}$ | $3.6036 \times 10^{-05}$ | $8.8161 \times 10^{-06}$ | $9.3734 \times 10^{-03}$ | $2.1106 \times 10^{-05}$ |
| | Mean | $8.8202 \times 10^{-05}$ | $2.1700 \times 10^{-04}$ | $2.0721 \times 10^{-04}$ | $3.7390 \times 10^{-04}$ | $1.8194 \times 10^{-02}$ | $1.7522 \times 10^{-02}$ |
| | Worst | $2.6899 \times 10^{-04}$ | $1.0933 \times 10^{-03}$ | $6.4167 \times 10^{-04}$ | $1.5364 \times 10^{-03}$ | $2.6649 \times 10^{-02}$ | $1.8120 \times 10^{-01}$ |
| | Std | $7.2479 \times 10^{-05}$ | $2.6392 \times 10^{-04}$ | $1.7316 \times 10^{-04}$ | $4.1133 \times 10^{-04}$ | $4.8117 \times 10^{-03}$ | $4.3922 \times 10^{-02}$ |

**Table 7.** Results of multi-modal benchmark functions.

| Func | Criteria | DQOBLSMA | OBLSMAL | ESMA | MEO | JADE | RWGWO |
|------|----------|----------|---------|------|-----|------|-------|
| F8 | Best | $-1.2569 \times 10^{+04}$ | $-8.8602 \times 10^{+03}$ | $-9.8908 \times 10^{+03}$ | $-5.4647 \times 10^{+03}$ | $-1.1856 \times 10^{+04}$ | $-9.3674 \times 10^{+03}$ |
| | Mean | $-1.2569 \times 10^{+04}$ | $-7.0233 \times 10^{+03}$ | $-8.5070 \times 10^{+03}$ | $-3.7623 \times 10^{+03}$ | $-1.0905 \times 10^{+04}$ | $-8.8801 \times 10^{+03}$ |
| | Worst | $-1.2569 \times 10^{+04}$ | $-5.4879 \times 10^{+03}$ | $-6.4963 \times 10^{+03}$ | $-3.0199 \times 10^{+03}$ | $-6.8045 \times 10^{+03}$ | $-8.0571 \times 10^{+03}$ |
| | Std | $1.0234 \times 10^{-01}$ | $7.7253 \times 10^{+02}$ | $8.5477 \times 10^{+02}$ | $5.5378 \times 10^{+02}$ | $1.6276 \times 10^{+03}$ | $3.2772 \times 10^{+02}$ |
| F9 | Best | $0.0000 \times 10^{+00}$ | $0.0000 \times 10^{+00}$ | $0.0000 \times 10^{+00}$ | $0.0000 \times 10^{+00}$ | $0.0000 \times 10^{+00}$ | $0.0000 \times 10^{+00}$ |
| | Mean | $0.0000 \times 10^{+00}$ | $0.0000 \times 10^{+00}$ | $0.0000 \times 10^{+00}$ | $0.0000 \times 10^{+00}$ | $9.4739 \times 10^{-15}$ | $0.0000 \times 10^{+00}$ |
| | Worst | $0.0000 \times 10^{+00}$ | $0.0000 \times 10^{+00}$ | $0.0000 \times 10^{+00}$ | $0.0000 \times 10^{+00}$ | $1.4744 \times 10^{-13}$ | $0.0000 \times 10^{+00}$ |
| | Std | $0.0000 \times 10^{+00}$ | $0.0000 \times 10^{+00}$ | $0.0000 \times 10^{+00}$ | $0.0000 \times 10^{+00}$ | $3.7368 \times 10^{-14}$ | $0.0000 \times 10^{+00}$ |
| F10 | Best | $4.4409 \times 10^{-16}$ | $4.4409 \times 10^{-16}$ | $4.4409 \times 10^{-16}$ | $4.4409 \times 10^{-16}$ | $7.7624 \times 10^{-08}$ | $4.4409 \times 10^{-16}$ |
| | Mean | $4.4409 \times 10^{-16}$ | $4.4409 \times 10^{-16}$ | $4.4409 \times 10^{-16}$ | $4.4409 \times 10^{-16}$ | $3.8505 \times 10^{-02}$ | $3.5231 \times 10^{-15}$ |
| | Worst | $4.4409 \times 10^{-16}$ | $4.4409 \times 10^{-16}$ | $4.4409 \times 10^{-16}$ | $4.4409 \times 10^{-16}$ | $1.1551 \times 10^{+00}$ | $3.9968 \times 10^{-15}$ |
| | Std | $0.0000 \times 10^{+00}$ | $0.0000 \times 10^{+00}$ | $0.0000 \times 10^{+00}$ | $0.0000 \times 10^{+00}$ | $2.7968 \times 10^{-01}$ | $1.2900 \times 10^{-15}$ |
| F11 | Best | $0.0000 \times 10^{+00}$ | $0.0000 \times 10^{+00}$ | $0.0000 \times 10^{+00}$ | $0.0000 \times 10^{+00}$ | $7.2387 \times 10^{-14}$ | $0.0000 \times 10^{+00}$ |
| | Mean | $0.0000 \times 10^{+00}$ | $0.0000 \times 10^{+00}$ | $0.0000 \times 10^{+00}$ | $0.0000 \times 10^{+00}$ | $4.3486 \times 10^{-03}$ | $0.0000 \times 10^{+00}$ |
| | Worst | $0.0000 \times 10^{+00}$ | $0.0000 \times 10^{+00}$ | $0.0000 \times 10^{+00}$ | $0.0000 \times 10^{+00}$ | $3.6770 \times 10^{-02}$ | $0.0000 \times 10^{+00}$ |
| | Std | $0.0000 \times 10^{+00}$ | $0.0000 \times 10^{+00}$ | $0.0000 \times 10^{+00}$ | $0.0000 \times 10^{+00}$ | $9.9297 \times 10^{-03}$ | $0.0000 \times 10^{+00}$ |
| F12 | Best | $5.8098 \times 10^{-08}$ | $1.7590 \times 10^{-02}$ | $2.8002 \times 10^{-02}$ | $1.1540 \times 10^{-02}$ | $4.4220 \times 10^{-14}$ | $2.9591 \times 10^{-02}$ |
| | Mean | $8.9207 \times 10^{-04}$ | $4.3823 \times 10^{-02}$ | $9.0114 \times 10^{-02}$ | $4.6612 \times 10^{-02}$ | $4.4934 \times 10^{-02}$ | $1.0348 \times 10^{-01}$ |
| | Worst | $3.6337 \times 10^{-03}$ | $1.2371 \times 10^{-01}$ | $4.2696 \times 10^{-01}$ | $8.6796 \times 10^{-02}$ | $4.1469 \times 10^{-01}$ | $7.3880 \times 10^{-01}$ |
| | Std | $1.0608 \times 10^{-03}$ | $2.4564 \times 10^{-02}$ | $1.0394 \times 10^{-01}$ | $2.0035 \times 10^{-02}$ | $1.2955 \times 10^{-01}$ | $1.6264 \times 10^{-01}$ |
| F13 | Best | $7.3229 \times 10^{-06}$ | $2.4407 \times 10^{-01}$ | $2.5338 \times 10^{-01}$ | $4.5254 \times 10^{-01}$ | $4.1497 \times 10^{-14}$ | $5.6485 \times 10^{-01}$ |
| | Mean | $1.4921 \times 10^{-03}$ | $1.0518 \times 10^{+00}$ | $7.9118 \times 10^{-01}$ | $8.9529 \times 10^{-01}$ | $2.3516 \times 10^{-10}$ | $1.1565 \times 10^{+00}$ |
| | Worst | $1.1660 \times 10^{-02}$ | $2.6596 \times 10^{+00}$ | $1.4767 \times 10^{+00}$ | $1.2682 \times 10^{+00}$ | $2.9604 \times 10^{-09}$ | $2.3763 \times 10^{+00}$ |
| | Std | $3.6813 \times 10^{-03}$ | $6.9507 \times 10^{-01}$ | $3.3716 \times 10^{-01}$ | $2.2110 \times 10^{-01}$ | $7.8166 \times 10^{-10}$ | $4.1169 \times 10^{-01}$ |

**Table 8.** Results of fixed-dimension multi-modal benchmark functions.

| Func | Criteria | DQOBLSMA | OBLSMAL | ESMA | MEO | JADE | RWGWO |
|------|----------|----------|---------|------|-----|------|-------|
| F14 | Best | $9.9800 \times 10^{-01}$ | $9.9800 \times 10^{-01}$ | $9.9800 \times 10^{-01}$ | $1.0937 \times 10^{+00}$ | $9.9800 \times 10^{-01}$ | $9.9800 \times 10^{-01}$ |
| | Mean | $9.9800 \times 10^{-01}$ | $1.1304 \times 10^{+00}$ | $1.0641 \times 10^{+00}$ | $5.7783 \times 10^{+00}$ | $9.9800 \times 10^{-01}$ | $1.7229 \times 10^{+00}$ |
| | Worst | $9.9800 \times 10^{-01}$ | $2.9821 \times 10^{+00}$ | $2.9821 \times 10^{+00}$ | $1.2671 \times 10^{+01}$ | $9.9800 \times 10^{-01}$ | $5.9288 \times 10^{+00}$ |
| | Std | $3.4807 \times 10^{-13}$ | $5.2272 \times 10^{-01}$ | $4.8038 \times 10^{-01}$ | $3.9055 \times 10^{+00}$ | $2.7756 \times 10^{-17}$ | $1.6192 \times 10^{+00}$ |
| F15 | Best | $3.0958 \times 10^{-04}$ | $3.0772 \times 10^{-04}$ | $5.8084 \times 10^{-04}$ | $3.0894 \times 10^{-04}$ | $3.0749 \times 10^{-04}$ | $4.1151 \times 10^{-04}$ |
| | Mean | $3.8029 \times 10^{-04}$ | $8.3277 \times 10^{-04}$ | $8.3114 \times 10^{-04}$ | $3.4423 \times 10^{-03}$ | $1.7361 \times 10^{-03}$ | $1.1214 \times 10^{-03}$ |
| | Worst | $6.3781 \times 10^{-04}$ | $1.2548 \times 10^{-03}$ | $1.2249 \times 10^{-03}$ | $2.0363 \times 10^{-02}$ | $2.0363 \times 10^{-02}$ | $2.6665 \times 10^{-03}$ |
| | Std | $9.0820 \times 10^{-05}$ | $3.3167 \times 10^{-04}$ | $2.1318 \times 10^{-04}$ | $7.2217 \times 10^{-03}$ | $5.8251 \times 10^{-03}$ | $5.9580 \times 10^{-04}$ |
| F16 | Best | $-1.0316 \times 10^{+00}$ | $-1.0316 \times 10^{+00}$ | $-1.0316 \times 10^{+00}$ | $-1.0316 \times 10^{+00}$ | $-1.0316 \times 10^{+00}$ | $-1.0316 \times 10^{+00}$ |
| | Mean | $-1.0316 \times 10^{+00}$ | $-1.0316 \times 10^{+00}$ | $-1.0316 \times 10^{+00}$ | $-1.0316 \times 10^{+00}$ | $-1.0316 \times 10^{+00}$ | $-1.0316 \times 10^{+00}$ |
| | Worst | $-1.0316 \times 10^{+00}$ | $-1.0316 \times 10^{+00}$ | $-1.0316 \times 10^{+00}$ | $-1.0316 \times 10^{+00}$ | $-1.0316 \times 10^{+00}$ | $-1.0307 \times 10^{+00}$ |
| | Std | $4.1555 \times 10^{-10}$ | $3.1460 \times 10^{-08}$ | $2.6995 \times 10^{-10}$ | $1.7352 \times 10^{-10}$ | $6.5564 \times 10^{-16}$ | $2.3316 \times 10^{-04}$ |
| F17 | Best | $3.9789 \times 10^{-01}$ | $3.9789 \times 10^{-01}$ | $3.9789 \times 10^{-01}$ | $3.9789 \times 10^{-01}$ | $3.9789 \times 10^{-01}$ | $3.9789 \times 10^{-01}$ |
| | Mean | $3.9789 \times 10^{-01}$ | $3.9789 \times 10^{-01}$ | $3.9789 \times 10^{-01}$ | $3.9789 \times 10^{-01}$ | $3.9789 \times 10^{-01}$ | $3.9792 \times 10^{-01}$ |
| | Worst | $3.9789 \times 10^{-01}$ | $3.9789 \times 10^{-01}$ | $3.9789 \times 10^{-01}$ | $3.9789 \times 10^{-01}$ | $3.9789 \times 10^{-01}$ | $3.9803 \times 10^{-01}$ |
| | Std | $3.2851 \times 10^{-08}$ | $1.4951 \times 10^{-07}$ | $3.3940 \times 10^{-08}$ | $5.0177 \times 10^{-09}$ | $0.0000 \times 10^{+00}$ | $4.2346 \times 10^{-05}$ |
| F18 | Best | $3.0000 \times 10^{+00}$ | $3.0000 \times 10^{+00}$ | $3.0000 \times 10^{+00}$ | $3.0000 \times 10^{+00}$ | $3.0000 \times 10^{+00}$ | $3.0000 \times 10^{+00}$ |
| | Mean | $3.0000 \times 10^{+00}$ | $3.0000 \times 10^{+00}$ | $3.0000 \times 10^{+00}$ | $3.0000 \times 10^{+00}$ | $3.0000 \times 10^{+00}$ | $3.0029 \times 10^{+00}$ |
| | Worst | $3.0000 \times 10^{+00}$ | $3.0000 \times 10^{+00}$ | $3.0000 \times 10^{+00}$ | $3.0000 \times 10^{+00}$ | $3.0000 \times 10^{+00}$ | $3.0315 \times 10^{+00}$ |
| | Std | $4.3415 \times 10^{-07}$ | $7.5551 \times 10^{-07}$ | $4.5784 \times 10^{-11}$ | $1.0614 \times 10^{-05}$ | $1.5740 \times 10^{-15}$ | $8.2802 \times 10^{-03}$ |
| F19 | Best | $-3.8628 \times 10^{+00}$ | $-3.8628 \times 10^{+00}$ | $-3.8628 \times 10^{+00}$ | $-3.8626 \times 10^{+00}$ | $-3.8628 \times 10^{+00}$ | $-3.8628 \times 10^{+00}$ |
| | Mean | $-3.8628 \times 10^{+00}$ | $-3.8628 \times 10^{+00}$ | $-3.8627 \times 10^{+00}$ | $-3.8589 \times 10^{+00}$ | $-3.8628 \times 10^{+00}$ | $-3.8520 \times 10^{+00}$ |
| | Worst | $-3.8628 \times 10^{+00}$ | $-3.8628 \times 10^{+00}$ | $-3.8616 \times 10^{+00}$ | $-3.8549 \times 10^{+00}$ | $-3.8628 \times 10^{+00}$ | $-3.7967 \times 10^{+00}$ |
| | Std | $6.0439 \times 10^{-07}$ | $7.0025 \times 10^{-06}$ | $2.8826 \times 10^{-04}$ | $2.7955 \times 10^{-03}$ | $2.6226 \times 10^{-15}$ | $1.7232 \times 10^{-02}$ |
| F20 | Best | $-3.3220 \times 10^{+00}$ | $-3.3220 \times 10^{+00}$ | $-3.3220 \times 10^{+00}$ | $-3.3220 \times 10^{+00}$ | $-3.3220 \times 10^{+00}$ | $-3.2948 \times 10^{+00}$ |
| | Mean | $-3.2821 \times 10^{+00}$ | $-3.2220 \times 10^{+00}$ | $-3.2313 \times 10^{+00}$ | $-3.2590 \times 10^{+00}$ | $-3.2903 \times 10^{+00}$ | $-3.1655 \times 10^{+00}$ |
| | Worst | $-3.1999 \times 10^{+00}$ | $-3.1985 \times 10^{+00}$ | $-3.0851 \times 10^{+00}$ | $-3.0633 \times 10^{+00}$ | $-3.2031 \times 10^{+00}$ | $-2.9487 \times 10^{+00}$ |
| | Std | $5.7002 \times 10^{-02}$ | $4.6895 \times 10^{-02}$ | $6.8225 \times 10^{-02}$ | $9.1478 \times 10^{-02}$ | $5.3456 \times 10^{-02}$ | $1.0946 \times 10^{-01}$ |
| F21 | Best | $-1.0153 \times 10^{+01}$ | $-1.0153 \times 10^{+01}$ | $-1.0153 \times 10^{+01}$ | $-5.1609 \times 10^{+00}$ | $-1.0153 \times 10^{+01}$ | $-1.0148 \times 10^{+01}$ |
| | Mean | $-1.0153 \times 10^{+01}$ | $-9.9934 \times 10^{+00}$ | $-9.3978 \times 10^{+00}$ | $-5.0587 \times 10^{+00}$ | $-9.3166 \times 10^{+00}$ | $-7.0389 \times 10^{+00}$ |
| | Worst | $-1.0152 \times 10^{+01}$ | $-7.5756 \times 10^{+00}$ | $-2.6300 \times 10^{+00}$ | $-5.0552 \times 10^{+00}$ | $-2.6305 \times 10^{+00}$ | $-5.0064 \times 10^{+00}$ |
| | Std | $2.1496 \times 10^{-04}$ | $6.7461 \times 10^{-01}$ | $2.4872 \times 10^{+00}$ | $2.5592 \times 10^{-02}$ | $2.4162 \times 10^{+00}$ | $2.4611 \times 10^{+00}$ |
| F22 | Best | $-1.0403 \times 10^{+01}$ | $-1.0403 \times 10^{+01}$ | $-1.0403 \times 10^{+01}$ | $-8.1136 \times 10^{+00}$ | $-1.0403 \times 10^{+01}$ | $-1.0391 \times 10^{+01}$ |
| | Mean | $-1.0403 \times 10^{+01}$ | $-9.1689 \times 10^{+00}$ | $-9.3977 \times 10^{+00}$ | $-5.2039 \times 10^{+00}$ | $-9.7170 \times 10^{+00}$ | $-7.0943 \times 10^{+00}$ |
| | Worst | $-1.0402 \times 10^{+01}$ | $-2.7484 \times 10^{+00}$ | $-2.7495 \times 10^{+00}$ | $-2.5429 \times 10^{+00}$ | $-2.7496 \times 10^{+00}$ | $-2.7426 \times 10^{+00}$ |
| | Std | $1.8317 \times 10^{-04}$ | $2.8605 \times 10^{+00}$ | $2.6374 \times 10^{+00}$ | $1.0562 \times 10^{+00}$ | $2.3627 \times 10^{+00}$ | $2.9401 \times 10^{+00}$ |
| F23 | Best | $-1.0536 \times 10^{+01}$ | $-1.0536 \times 10^{+01}$ | $-1.0536 \times 10^{+01}$ | $-1.0536 \times 10^{+01}$ | $-1.0536 \times 10^{+01}$ | $-1.0536 \times 10^{+01}$ |
| | Mean | $-1.0536 \times 10^{+01}$ | $-8.8867 \times 10^{+00}$ | $-9.2412 \times 10^{+00}$ | $-7.4582 \times 10^{+00}$ | $-1.0536 \times 10^{+01}$ | $-6.4373 \times 10^{+00}$ |
| | Worst | $-1.0536 \times 10^{+01}$ | $-2.4177 \times 10^{+00}$ | $-2.4216 \times 10^{+00}$ | $-5.1285 \times 10^{+00}$ | $-1.0536 \times 10^{+01}$ | $-2.4270 \times 10^{+00}$ |
| | Std | $2.0415 \times 10^{-04}$ | $3.1383 \times 10^{+00}$ | $3.0570 \times 10^{+00}$ | $2.5657 \times 10^{+00}$ | $1.9610 \times 10^{-15}$ | $2.5968 \times 10^{+00}$ |

**Table 9.** Test statistical results of Wilcoxon's rank-sum test.

| Benchmark | DQOBLSMA vs. OBLSMAL | | DQOBLSMA vs. ESMA | | DQOBLSMA vs. MEO | | DQOBLSMA vs. JADE | | DQOBLSMA vs. RWGWO | |
|---|---|---|---|---|---|---|---|---|---|---|
| | *p*-Value | Winner | *p*-Value | Winner | *p*-Value | Winner | *p*-Value | Winner | *p*-Value | Winner |
| F1 | NaN | = | NaN | = | $1.73 \times 10^{-06}$ | + | $1.73 \times 10^{-06}$ | + | $1.73 \times 10^{-06}$ | + |
| F2 | $1.73 \times 10^{-06}$ | + | $1.73 \times 10^{-06}$ | + | $1.73 \times 10^{-06}$ | + | $1.73 \times 10^{-06}$ | + | $1.73 \times 10^{-06}$ | + |
| F3 | $1.73 \times 10^{-06}$ | + | $1.73 \times 10^{-06}$ | + | $1.73 \times 10^{-06}$ | + | $1.73 \times 10^{-06}$ | + | $1.73 \times 10^{-06}$ | + |
| F4 | $1.73 \times 10^{-06}$ | + | $1.73 \times 10^{-06}$ | + | $1.73 \times 10^{-06}$ | + | $1.73 \times 10^{-06}$ | + | $1.73 \times 10^{-06}$ | + |
| F5 | $1.73 \times 10^{-06}$ | + | $1.73 \times 10^{-06}$ | + | $7.04 \times 10^{-01}$ | + | $1.73 \times 10^{-06}$ | + | $1.73 \times 10^{-06}$ | + |
| F6 | NaN | = | NaN | = | NaN | = | NaN | + | NaN | = |
| F7 | $2.41 \times 10^{-03}$ | + | $4.20 \times 10^{-04}$ | + | $1.73 \times 10^{-06}$ | + | $1.73 \times 10^{-06}$ | + | $9.32 \times 10^{-06}$ | + |
| F8 | $1.73 \times 10^{-06}$ | + | $1.73 \times 10^{-06}$ | + | $1.73 \times 10^{-06}$ | + | $1.73 \times 10^{-06}$ | + | $1.73 \times 10^{-06}$ | + |
| F9 | NaN | = | NaN | = | NaN | = | $1.20 \times 10^{-02}$ | + | NaN | = |
| F10 | NaN | = | NaN | = | NaN | = | $1.73 \times 10^{-06}$ | + | $6.39 \times 10^{-07}$ | + |
| F11 | NaN | = | NaN | = | NaN | = | $1.73 \times 10^{-06}$ | + | NaN | = |
| F12 | $1.73 \times 10^{-06}$ | + | $1.73 \times 10^{-06}$ | + | $2.61 \times 10^{-04}$ | + | NaN | + | $1.73 \times 10^{-06}$ | + |
| F13 | $1.73 \times 10^{-06}$ | + | $1.73 \times 10^{-06}$ | + | $2.07 \times 10^{-02}$ | + | $1.73 \times 10^{-06}$ | − | $1.73 \times 10^{-06}$ | + |
| F14 | $2.61 \times 10^{-04}$ | + | $2.71 \times 10^{-01}$ | + | $1.73 \times 10^{-06}$ | + | $1.73 \times 10^{-06}$ | − | $1.73 \times 10^{-06}$ | + |
| F15 | $6.34 \times 10^{-06}$ | + | $1.73 \times 10^{-06}$ | + | $3.00 \times 10^{-02}$ | + | NaN | + | $1.73 \times 10^{-06}$ | + |
| F16 | $4.29 \times 10^{-06}$ | = | $1.17 \times 10^{-02}$ | − | $4.73 \times 10^{-06}$ | − | $1.73 \times 10^{-06}$ | = | $1.73 \times 10^{-06}$ | = |
| F17 | $7.51 \times 10^{-05}$ | = | NaN | = | $4.86 \times 10^{-05}$ | − | $1.73 \times 10^{-06}$ | = | $1.73 \times 10^{-06}$ | − |
| F18 | $3.88 \times 10^{-04}$ | = | $9.32 \times 10^{-06}$ | − | $5.75 \times 10^{-06}$ | = | $1.73 \times 10^{-06}$ | − | $1.73 \times 10^{-06}$ | + |
| F19 | $7.71 \times 10^{-04}$ | = | $1.74 \times 10^{-04}$ | + | $1.73 \times 10^{-06}$ | + | $1.73 \times 10^{-06}$ | − | $1.73 \times 10^{-06}$ | + |
| F20 | $1.89 \times 10^{-04}$ | + | $8.94 \times 10^{-04}$ | + | $1.73 \times 10^{-06}$ | + | $1.75 \times 10^{-02}$ | − | $1.74 \times 10^{-04}$ | + |
| F21 | $1.73 \times 10^{-06}$ | + | $4.29 \times 10^{-06}$ | + | $1.73 \times 10^{-06}$ | + | $1.48 \times 10^{-02}$ | + | $1.73 \times 10^{-06}$ | + |
| F22 | $5.22 \times 10^{-06}$ | + | $1.02 \times 10^{-05}$ | + | $1.73 \times 10^{-06}$ | + | $2.77 \times 10^{-03}$ | + | $1.73 \times 10^{-06}$ | + |
| F23 | $5.22 \times 10^{-06}$ | + | $1.38 \times 10^{-03}$ | + | $1.73 \times 10^{-06}$ | + | $1.73 \times 10^{-06}$ | − | $1.73 \times 10^{-06}$ | + |
| +/−/= | 14/0/9 | | 15/2/6 | | 16/2/5 | | 15/6/2 | | 18/1/4 | |

**Table 10.** Test statistical results of the Friedman test.

| Func | DQOBLSMA | OBLSMAL | ESMA | MEO | JADE | RWGWO |
|------|----------|---------|------|-----|------|-------|
| F1–F7 | 1.36 | 3 | 2.36 | 3.86 | 5.71 | 4.71 |
| F8–F13 | 2.08 | 3.42 | 3.42 | 3.75 | 4 | 4.33 |
| F14–23 | 2.45 | 3.7 | 3 | 4.5 | 1.85 | 5.5 |
| F1–F23 | 2.02 | 3.41 | 2.91 | 4.11 | 3.59 | 4.96 |

*4.4. Convergence Analysis*

To demonstrate the effectiveness of the proposed DQOBLSMA, Figure 2 shows the convergence curves of the DQOBLSMA, SMA, ESMA, AEO, JADE, and RWGWO for the classical benchmark functions. The convergence curves show that the initial convergence of the DQOBLSMA was the fastest in most cases, except for $F6$, $F9$, $F10$, and $F11$; and RWGWO had faster initial convergence for these functions. For $F16$–$F20$, all comparison algorithms converged quickly to the global optimum, and the DQOBLSMA did not show a significant advantage. In Figure 2, a step or cliff drop in the DQOBLSMA's convergence curve can be observed, which indicates outstanding exploration capability. In almost all test cases, the DQOBLSMA had a better convergence rate than SMA and SMA variants, indicating that the SMA's convergence results can be significantly improved when applying the proposed search strategies. In conclusion, the DQOBLSMA is not only robust and effective at producing the best results, but also has a higher convergence speed than the other algorithms.

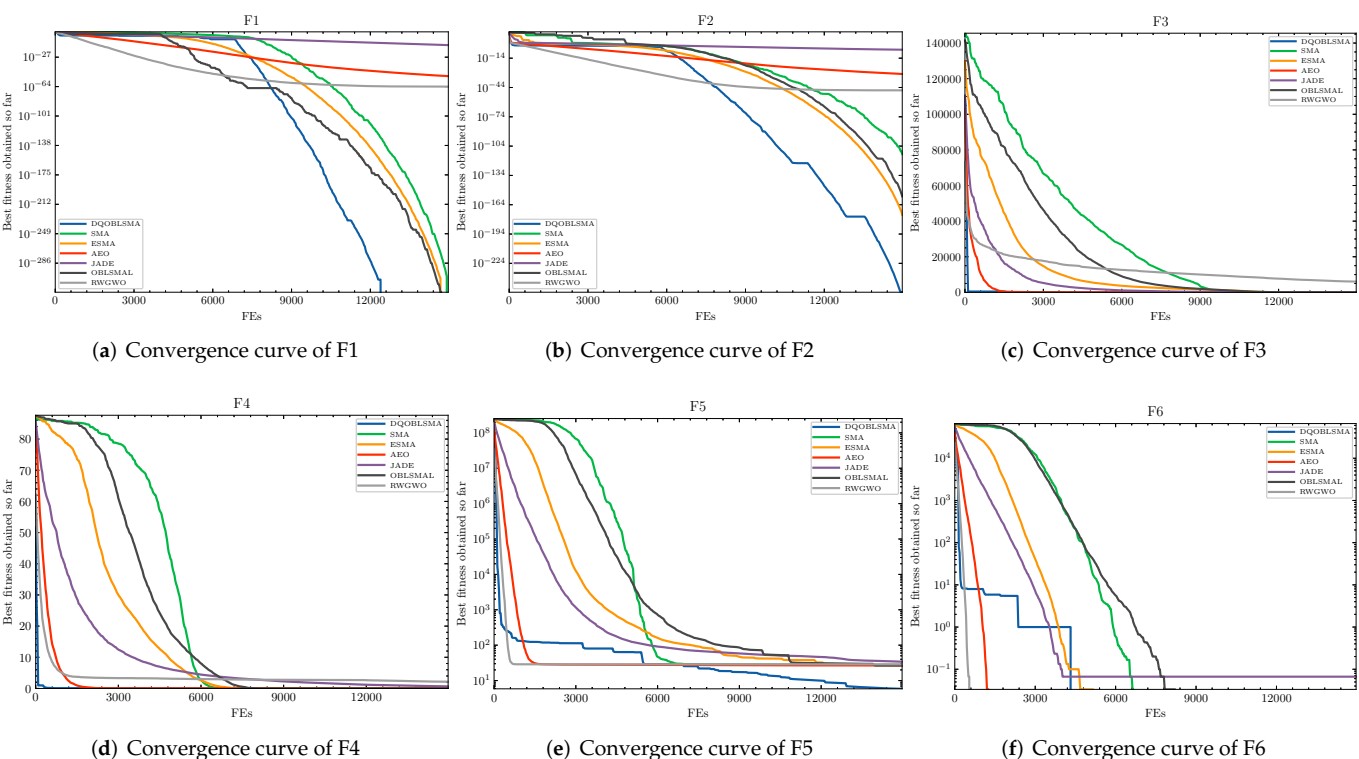

(**a**) Convergence curve of F1  (**b**) Convergence curve of F2  (**c**) Convergence curve of F3

(**d**) Convergence curve of F4  (**e**) Convergence curve of F5  (**f**) Convergence curve of F6

**Figure 2.** *Cont.*

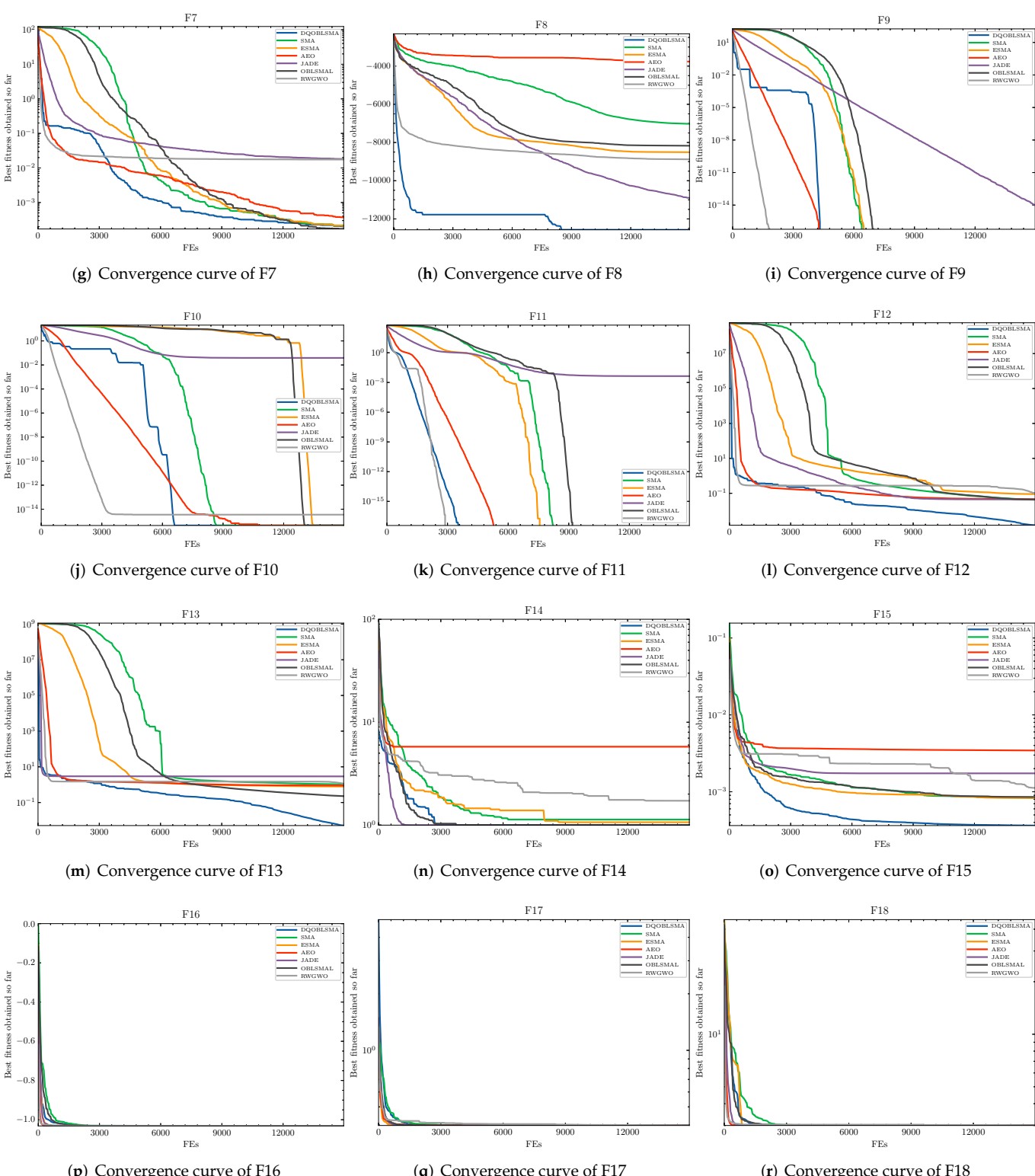

(**g**) Convergence curve of F7

(**h**) Convergence curve of F8

(**i**) Convergence curve of F9

(**j**) Convergence curve of F10

(**k**) Convergence curve of F11

(**l**) Convergence curve of F12

(**m**) Convergence curve of F13

(**n**) Convergence curve of F14

(**o**) Convergence curve of F15

(**p**) Convergence curve of F16

(**q**) Convergence curve of F17

(**r**) Convergence curve of F18

**Figure 2.** *Cont.*

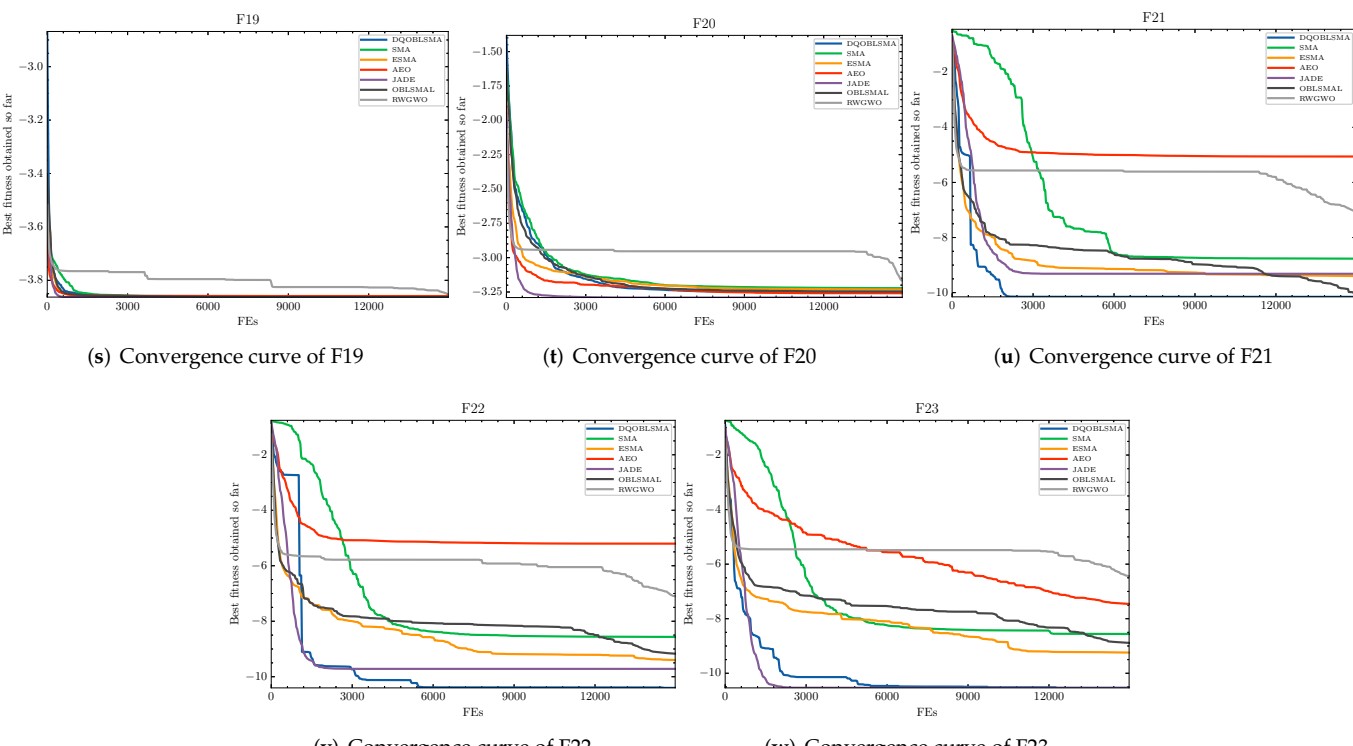

(**s**) Convergence curve of F19     (**t**) Convergence curve of F20     (**u**) Convergence curve of F21

(**v**) Convergence curve of F22     (**w**) Convergence curve of F23

**Figure 2.** Convergence figures on test functions F1–F23.

## 5. Engineering Design Problems

In this section, the DQOBLSMA is evaluated using three engineering design problems: the welded beam design problem, tension/compression springs, and the pressure vessel design problem. These engineering problems are well known and have been widely used to verify the effectiveness of methods for solving complex real-world problems [54]. The proposed method is compared with the state-of-the-art algorithms: OBLSMAL, ESMA, MEO, JADE, and RWGWO. The population size (N) and the maximum number of iterations were fixed at 30 and 500 for all comparison algorithms.

### 5.1. Welded Beam Design Problem

The design diagram for the structural problem of a welded beam [55] is shown in Figure 3. The objective of structural design optimization of welded beams is to minimize the total cost, subject to certain constraints, which are the shear stress $\tau$, the bending stress $\sigma$ on the beam, the buckling load $P_c$, and the deflection $\delta$ of beam. Four variables are considered in this problem, welded thickness ($h$), the bar length ($l$), bar height ($t$), and the thickness of the bar ($b$).

The mathematical equations of this problem are shown below:
Consider:

$$\mathbf{x} = [x_1 \ x_2 \ x_3 \ x_4] = [h \ l \ t \ b];$$

minimize:

$$f(\mathbf{x}) = 1.10471x_1^2x_2 + 0.04811x_3x_4(14 + x_2);$$

subject to:

$$g_1(\mathbf{x}) = \sqrt{(\tau')^2 + 2\tau'\tau''\frac{x_2}{2R} + (\tau'')^2} - \tau_{max} \leq 0;$$

$$g_2(\mathbf{x}) = \frac{6PL}{x_3^2 x_4} - \sigma_{max} \leq 0;$$

$$g_3(\mathbf{x}) = x_1 - x_4 \leq 0;$$

$$g_4(\mathbf{x}) = 0.10471x_1^2 + 0.04811x_3x_4(14 + x_2) - 5 \leq 0;$$

$$g_5(\mathbf{x}) = 0.125 - x_1 \leq 0;$$

$$g_6(\mathbf{x}) = \frac{4PL^3}{Ex_3^3 x_4} - \delta_{max} \leq 0;$$

$$g_7(\mathbf{x}) = P - \frac{4.013Ex_3x_4^3}{6L^2}\left(1 - \frac{x_3}{2L}\sqrt{\frac{E}{4G}}\right) \leq 0;$$

where:

$$\tau' = \frac{P}{2x_1x_2}, \tau'' = MRJ, M = P(L + \frac{x_2}{2}),$$

$$J = 2\left\{\sqrt{2}x_1x_2\left[\frac{x_2^2}{12} + \left(\frac{x_1 + x_3}{2}\right)^2\right]\right\},$$

$$R = \sqrt{\frac{x_2^2}{4} + \left(\frac{x_1 + x_3}{2}\right)^2}, P = 6000lb,$$

$$L = 14in, E = 30 \times 10^6 psi, G = 12 \times 10^6 psi,$$

$$\tau_{max} = 13600psi, \sigma_{max} = 30000psi,$$

$$\delta_{max} = 0.25in;$$

range of variables:

$$0.1 \leq x_1, x_4 \leq 2.0 \text{ and } 0.1 \leq x_2, x_3 \leq 10.0$$

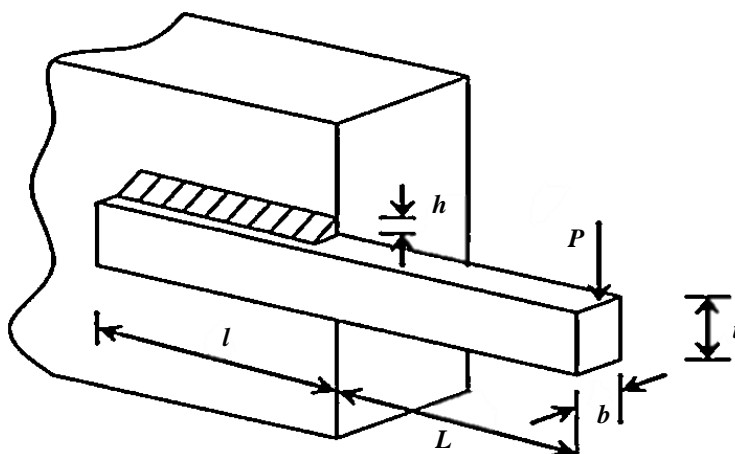

**Figure 3.** Welded beam design problem.

In Table 11, the results of the proposed DQOBLSMA and other well-known comparative optimization algorithms are given. It is clear from Table 11 that the proposed DQOBLSMA provides promising results for the optimal variables compared to other well-known optimization algorithms. The DQOBLSMA obtained a minimum cost of 1.695436 when $h$ = 0.205598, $l$ = 3.255605 , $t$ = 9.036367, and $b$ = 0.205741.

**Table 11.** Comparison in welded beam design.

| Algorithm | Optimal Values for Variables | | | | Optimal Cost |
|---|---|---|---|---|---|
| | $h$ | $l$ | $t$ | $b$ | |
| DQOBLSMA | 0.205598 | 3.255605 | 9.036367 | 0.205741 | 1.695436 |
| OBLSMAL | 0.253062 | 1.842203 | 8.270240 | 0.253229 | 1.726511 |
| ESMA | 0.201567 | 3.357515 | 8.983361 | 0.208407 | 1.712227 |
| SMA | 0.197433 | 3.407377 | 9.036868 | 0.205729 | 1.703704 |
| MEO | 0.194411 | 3.487386 | 9.040436 | 0.205984 | 1.712024 |
| JADE | 0.205734 | 3.253036 | 9.036624 | 0.205730 | 1.695245 |
| RWGWO | 0.247585 | 3.000055 | 8.090046 | 0.256700 | 1.901643 |

*5.2. Tension/Compression Spring Design*

The design goal for extension/compression springs [56] is to obtain the minimum optimum weight under four constraints: deviation ($g_1$), shear stress ($g_2$), surge frequency ($g_3$), and deflection ($g_4$). As shown in Figure 4, three variables need to be considered. They are the wire diameter ($d$), the mean coil diameter ($D$), and the number of active coils ($N$). The mathematical description of this problem is given below:

Consider:

$$\mathbf{x} = [x_1\ x_2\ x_3] = [d\ D\ N];$$

minimize:

$$f(\mathbf{x}) = x_1^2 x_2 (2 + x_3);$$

subject to:

$$g_1(\mathbf{x}) = 1 - \frac{x_2^3 x_3}{71785 x_1^4} \leq 0;$$

$$g_2(\mathbf{x}) = \frac{4x_2^2 - x_1 x_2}{12566(x_2 x_1^3 - x_1^4)} + \frac{1}{5108 x_1^2} \leq 0;$$

$$g_3(\mathbf{x}) = 1 - \frac{140.45 x_1}{x_2^2 x_3} \leq 0;$$

$$g_4(\mathbf{x}) = \frac{x_1 + x_2}{1.5} - 1 \leq 0;$$

range of variables:

$$0.05 \leq x_1 \leq 2.0, 0.25 \leq x_2 \leq 1.3, and\ 2.0 \leq x_3 \leq 15.0.$$

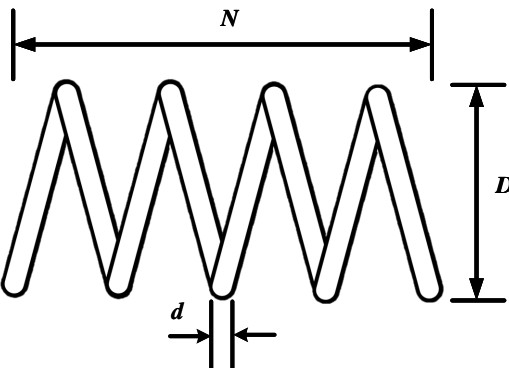

**Figure 4.** Tension/compression spring design problem.

The results of the DQOBLSMA and other comparative algorithms are presented in Table 12. The proposed DQOBLSMA achieved the best solution to the problem. The DQOBLSMA obtained a minimum cost of 0.012719 when $d$ = 0.050000, $D$ = 0.317425, and $N$ = 14.028013.

**Table 12.** Comparison for the tension/compression spring design problem.

| Algorithm | Optimal Values for Variables | | | Optimal Cost |
|-----------|------|------|------|--------------|
| | $d$ | $D$ | $N$ | |
| DQOBLSMA | 0.050000 | 0.317425 | 14.028013 | 0.012719 |
| OBLSMAL | 0.050000 | 0.317409 | 14.030650 | 0.012721 |
| ESMA | 0.051458 | 0.353086 | 12.050995 | 0.012739 |
| SMA | 0.050000 | 0.317317 | 14.042338 | 0.012726 |
| MEO | 0.057203 | 0.514683 | 7.661607 | 0.014002 |
| JADE | 0.055015 | 0.442128 | 7.613118 | 0.012864 |
| RWGWO | 0.056389 | 0.480684 | 6.712235 | 0.013316 |

*5.3. Pressure Vessel Design*

The pressure vessel design problem is a four-variable, four-constraint problem in the industry field that aims to reduce the total cost of a given cylindrical pressure vessel [57]. The four variables studied include the width of the shell ($Ts$), the width of the head ($Th$), the inner radius ($R$), and the length of the cylindrical section ($L$), as shown in Figure 5. The formulation of objective functions and four optimization constraints can be described as follows:

Consider:

$$\mathbf{x} = [x_1\ x_2\ x_3\ x_4] = [T_s\ T_h\ R\ L];$$

minimize:

$$f(x) = 0.6224x_1 x_3 x_4 + 1.7781x_2 x_3^2 + 3.1661x_1^2 x_4 + 19.84x_1^2 x_3;$$

subject to:

$$g_1(x) = -x_1 + 0.0193x_3 \leq 0$$
$$g_2(x) = -x_3 + 0.00954x_3 \leq 0$$
$$g_3(x) = -\pi x_3^2 x_4 - \frac{4}{3}\pi x_3^3 + 1296000 \leq 0$$
$$g_4(x) = x_4 - 240 \leq 0;$$

range of variables:

$$0 \le x_1 \le 99, 0 \le x_2 \le 99, 10 \le x_3 \le 200, 10 \le x_4 \le 200$$

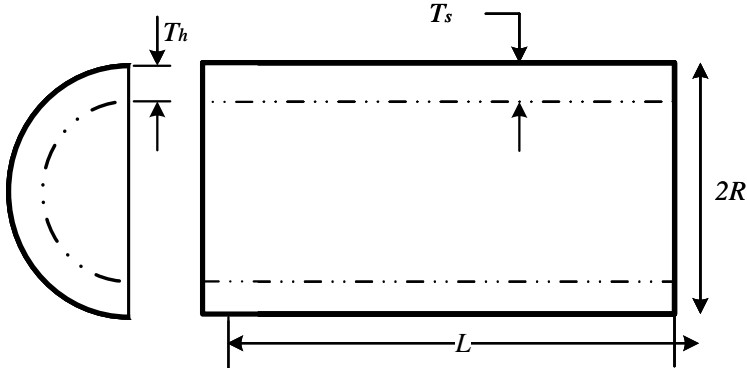

**Figure 5.** Pressure vessel design problem.

Table 13 shows how the DQOBLSMA compares with other competitor algorithms. The results shows the DQOBLSMA is able to find the optimal solution at the lowest cost, obtaining an optimal spend of 5885.623524 when $Ts = 0.778246$, $Th = 0.384708$, $R = 40.323469$, and $L = 199.950065$.

**Table 13.** Comparison in pressure vessel design.

| Algorithm | Optimal Values for Variables | | | | Optimal Cost |
|---|---|---|---|---|---|
| | Ts | Th | R | L | |
| DQOBLSMA | 0.778246 | 0.384708 | 40.323469 | 199.950065 | 5885.623524 |
| OBLSMAL | 0.865273 | 0.427877 | 44.832637 | 145.769573 | 6060.212044 |
| ESMA | 0.974581 | 0.481740 | 50.496415 | 112.689545 | 6417.418230 |
| SMA | 0.814081 | 0.402437 | 42.180339 | 175.629283 | 5949.827184 |
| MEO | 0.850407 | 0.425437 | 44.051816 | 154.133369 | 6046.777664 |
| JADE | 0.788821 | 0.389961 | 40.870447 | 192.471633 | 5904.076066 |
| RWGWO | 0.877511 | 0.432390 | 45.308765 | 140.703767 | 6095.405916 |

## 6. Conclusions

In this paper, an enhanced SMA (DQOBLSMA) was proposed by introducing two mechanisms, DQRG and OBL, into the original SMA. In the DQOBLSMA, these two strategies further enhance the global search capability of the original SMA: DQRG enhances the exploration capability of the original SMA, and OBL increases the population diversity. The DQOBLSMA overcomes the weaknesses of the original search method and avoids premature convergence. The performance of the proposed DQOBLSMA was analyzed by using 23 classical mathematical benchmark functions.

First, the DQOBLSMA and the individual combinations of these two strategies were analyzed and discussed. The results showed that the proposed strategies are effective, and SMA achieved the best performance with the combination of the two mechanisms. Secondly, the results of the DQOBLSMA were compared with five state-of-the-art algorithms ESMA, AEO, JADE, OBLSMAL, and RWGWO. The results show that the DQOBLSMA is competitive with other advanced metaheuristic algorithms. To further validate the superiority of the DQOBLSMA, it was applied to three industrial engineering design problems. The experimental results show that the DQOBLSMA also achieves better results when solving engineering problems and significantly improves the original solutions.

As a future perspective, a multi-objective version of the DQOBLSMA will be considered. The proposed algorithm has promising applications in scheduling problems, image

segmentation, parameter estimation, multi-objective engineering problems, text clustering, feature selection, test classification, and web applications.

**Author Contributions:** Conceptualization, S.D. and Y.Z.; software, Y.Z.; validation, S.D. and Q.Z.; formal analysis, S.D. and Y.Z.; investigation, S.D. and Y.Z.; resources, S.D.; writing—original draft preparation, Y.Z.; writing—review and editing, S.D. and Y.Z.; visualization, Y.Z.; funding acquisition, S.D. All authors have read and agreed to the published version of the manuscript.

**Funding:** The authors acknowledge the support of the Key R & D Projects of Zhejiang Province (No. 2022C01236, 2019C01060), the National Natural Science Foundations of China (Grant Nos. 21875271, U20B2021, 21707147, 51372046, 51479037, 91226202, and 91426304), the Entrepreneurship Program of Foshan National Hi-tech Industrial Development Zone, the Major Project of the Ministry of Science and Technology of China (Grant No. 2015ZX06004-001), Ningbo Natural Science Foundations (Grant Nos. 2014A610006, 2016A610273, and 2019A610106).

**Institutional Review Board Statement:** Not applicable.

**Informed Consent Statement:** Not applicable.

**Data Availability Statement:** Not applicable.

**Conflicts of Interest:** The authors declare no conflict of interest.

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
