# Peer review of "Improved Slime Mold Algorithm with Dynamic Quantum Rotation Gate and Opposition-Based Learning for Global Optimization and Engineering Design Problems"

_algorithms, doi:10.3390/a15090317_

Round 1

Reviewer 1 Report

The manuscript is well written and organized. However I would argue the gain in comparison to traditional SMA is marginally. It seems to be merely a combination of three different algorithms together, which does not include enough innovation. The author need to focus on what significantly distinguish this proposed algorithm over the other algorithm. maybe in a more specific area

Author Response

Response to Reviewer 1 Comments

Point 1: The manuscript is well written and organized. However I would argue the gain in comparison to traditional SMA is marginally.

Response 1:  Dear reviewer, thank you very much for your comments. The original SMA has a tendency to develop and explore unbalanced and fall into local optimality when solving complex problems. To overcome these drawbacks, an improved SMA with a dynamic quantum rotation gate and opposition-based learning(DQOBLSMA) is proposed in this paper.

The performance of the proposed DQOBLSMA is analyzed by using 23 classical mathematical benchmark functions. Experimental results show that the proposed algorithm outperforms SMA in convergence speed, convergence accuracy, and reliability.Thanks again.

Point 2: It seems to be merely a combination of three different algorithms together, which does not include enough innovation.The author need to focus on what significantly distinguish this proposed algorithm over the other algorithm. maybe in a more specific area

Response 2: Dear reviewer, thank you very much for your suggestions. In DQOBLSMA, these two strategies jointly improve the global search capability of the original SMA, in which DQRG enhances the exploration capability of the original SMA, and OBL increases the population diversity. The improvement is achieved by the joint effect instead of simple addition. In this paper, DQRG is proposed for the first time to overcome the problem that the original QRG method cannot balance exploration and exploitation. For the first time, both DQRG and OBL are introduced simultaneously into SMA to improve the algorithm performance. We plan to use the proposed algorithm for feature selection to improve the accuracy.

Thanks again.

Reviewer 2 Report

The paper proposes an improved version of the Slime Mould metaheuristic algorithm (SMA). As it is known that metaheuristics trade the accuracy of the results for speed and algorithm simplicity, research on the improvement of existing methods is an ongoing effort, with high relevance.

The approach proposed by the authors here has several strong points, as follows:

- a bidirectional improvement method is proposed, and the results show that indeed both changes have a contribution in improving the results of the original algorithm. It is clear what was attempted.

- for result validation, both theoretical and practical engineering problems are used

- the results section is extensive, and it presents information that prove with multiple tests the improvement of the original algorithm

However, the implementation presented in the paper shows some significant drawbacks:

Main concerns:

- As it is stated at page 2, line 54, several attempts to improve the original SMA algorithms are already proposed. But the authors chose to apply their improvement technique on the original SMA, as stated at page 2, line 83. A more logical approach would be to build upon the previous improvements. While performance comparisons are provided for other metaheuristics, no checks are made against other improved SMA implementations. This would be useful for evaluating the true performance of the authors' work.

- Section 3.4 (computational complexity analysis) is very brief. It would be more illustrative to make a comparison with the other metaheuristics or SMA implementations.

- Table 5 shows the running parameter setup for each metaheuristic used for comparison. As it is known that the performance of metaheuristic algorithms is heavily dependent on these parameters, the question arises if the values presented by the authors are optimized. More details should begiven on this matter.

- The engineering problems are only briefly stated, as mathematical equations. The objectives and especially the constraints should be explained more thoroughly.

- If Tables 5, 11 and 13 are analyzed, they show that the number and names of algorithms used for comparison varies. GA appears only in Tables 11 and 13. For a valid comparison, the same algorithms should be used in all subsections.

Also, some small corrections are needed:

- page 2, line 53: there is an unfinished sentence.

- page 18, line 275: Figure 2 is referenced as Figure 6.

- in Section 5, at page 20, line 287, it is said that five engineering problems are used for testing, while the case study presents only three.

Author Response

Response to Reviewer 2 Comments

Point 1: As it is stated at page 2, line 54, several attempts to improve the original SMA algorithms are already proposed. But the authors chose to apply their improvement technique on the original SMA, as stated at page 2, line 83. A more logical approach would be to build upon the previous improvements.

Response 1: Dear reviewer, thank you very much for your suggestions. OBL is a widely used improvement strategy for metaheuristic algorithms[1-3], which has been used to improve SMA but still cannot perfectly solve the problems suffered by SMA[4]. Meanwhile, QGR is an effective improvement strategy that has been applied to improve moth flame optimization[5] and hunger games search[6]. In this paper, we propose a modified QRG (DQGR), then both OBL and DQRG strategies are applied to SMA simultaneously to obtain the best boosting effect, and experiments are conducted in Section 4.2 to analyze the effects of different strategies. In other words, the improvement is achieved by the joint effect instead of simple addition. Thanks again.

Point 2: While performance comparisons are provided for other metaheuristics, no checks are made against other improved SMA implementations. This would be useful for evaluating the true performance of the authors' work.

Response 2: Dear reviewer, thank you very much for your suggestions. We follow your suggestion. We have added a comparison of DQOBLSMA with other improved SMAs and advanced algorithms: improved slime mould algorithm by opposition-based learning and Levy flight distribution(OBLSMAL), equilibrium slime mould algorithm (ESMA), equilibrium optimizer with mutation strategy(MEO), adaptive differential evolution with optional external archive(JADE), and grey wolf optimizer based on random walk(RWGWO). Thanks again.

Point 3: Section 3.4 (computational complexity analysis) is very brief. It would be more illustrative to make a comparison with the other metaheuristics or SMA implementations.

Response 3: Dear reviewer, thank you very much for your suggestions. We follow your suggestion. We added an analysis of the computational complexity of the original SMA, through which the improved strategy proposed in this paper does not increase the computational complexity compared with the original SMA. Thanks again.

Point 4: Table 5 shows the running parameter setup for each metaheuristic used for comparison. As it is known that the performance of metaheuristic algorithms is heavily dependent on these parameters, the question arises if the values presented by the authors are optimized. More details should begiven on this matter.

Response 4: Dear reviewer, thank you very much for your suggestions. We follow your suggestion. We present the sources of the comparison algorithms in Section 4.1, and the experimental parameters for all optimization algorithms are chosen to be the same as those reported in the original work. This verifies the improvement in methodology under the same condition. Optimization of the parameters depends on specific problems and thus could be carried out in future work. Moreover, in order to maintain fair comparison, each algorithm is independently run 30 times, and the population size (N) and the maximum function evaluation times (FEs) of all experimental methods are fixed at 30 and 15000. Thanks again.   

Point 5: The engineering problems are only briefly stated, as mathematical equations. The objectives and especially the constraints should be explained more thoroughly.

Response 5: Dear reviewer, thank you very much for your suggestions. The three engineering problems in the paper are widely used to test the performance of metaheuristic algorithms. They are abstracted into definite mathematical formulations making it easy for researchers to test the ability of the algorithms to solve real engineering problems with the same conditions. We have described the physical meanings, variables and constraints of the engineering problem in more detail in the revised paper. Thanks again.

Point 6: If Tables 5, 11 and 13 are analyzed, they show that the number and names of algorithms used for comparison varies. GA appears only in Tables 11 and 13. For a valid comparison, the same algorithms should be used in all subsections.

Response 6: Dear reviewer, thank you very much for your suggestions. We follow your suggestion. We used the same algorithm for comparison in all comparison experiments. The parameters of all comparison algorithms are described in detail in Table 5. Thanks again.

Point 7: Also, some small corrections are needed:- page 2, line 53: there is an unfinished sentence.- page 18, line 275: Figure 2 is referenced as Figure 6.- in Section 5, at page 20, line 287, it is said that five engineering problems are used for testing, while the case study presents only three.

Response 7: Dear reviewer, thank you very much for your suggestions. According to your revision comments, the above error has been corrected. Thanks again.

References

[1]Tubishat M, Idris N, Shuib L, Abushariah MA, Mirjalili S (2020) Improved salp swarm algorithm based on opposition based learning and novel local search algorithm for feature selection. Expert Syst Appl 145:113122

[2]Eltamaly A M, Al-Saud M, Sayed K, et al. Sensorless active and reactive control for DFIG wind turbines using opposition-based learning technique[J]. Sustainability, 2020, 12(9): 3583.

[3]Ewees A A, Abd Elaziz M, Houssein E H. Improved grasshopper optimization algorithm using opposition-based learning[J]. Expert Systems with Applications, 2018, 112: 156-172.

[4]Abualigah L, Diabat A, Elaziz M A. Improved slime mould algorithm by opposition-based learning and Levy flight distribution for global optimization and advances in real-world engineering problems[J]. Journal of Ambient Intelligence and Humanized Computing, 2021: 1-40.

[5]Yu C, Heidari A A, Chen H. A quantum-behaved simulated annealing algorithm-based moth-flame optimization method[J]. Applied Mathematical Modelling, 2020, 87: 1-19.

[6]Xu B, Heidari A A, Kuang F, et al. Quantum Nelder‐Mead Hunger Games Search for optimizing photovoltaic solar cells[J]. International Journal of Energy Research, 2022.

Round 2

Reviewer 1 Report

Thanks for clarifying my previous comments.

Reviewer 2 Report

The reviewer thanks the authors for the changes made to the paper.

The comments were addressed and the appropriate modifications were made.

The issues were largely corrected, and the paper can now be published.

Good luck!